# Mechano-boosting nanomedicine anti-tumour efficacy by blocking the reticuloendothelial system with stiff nanogels

Zheng Li[1], Yabo Zhu[1], Haowen Zeng[1], Chong Wang[1], Chen Xu[1], Qiang Wang[1], Huimin Wang[1], Shiyou Li[1], Jitang Chen[1], Chen Xiao[1], Xiangliang Yang [1,2,3,4] & Zifu Li [1,2,3,5] ✉

Nanomedicine has been developed for cancer therapy over several decades, while rapid clearance from blood circulation by reticuloendothelial system (RES) severely limits nanomedicine antitumour efficacy. We design a series of nanogels with distinctive stiffness and investigate how nanogel mechanical properties could be leveraged to overcome RES. Stiff nanogels are injected preferentially to abrogate uptake capacity of macrophages and temporarily block RES, relying on inhibition of clathrin and prolonged liver retention. Afterwards, soft nanogels deliver doxorubicin (DOX) with excellent efficiency, reflected in high tumour accumulation, deep tumour penetration and outstanding antitumour efficacy. In this work, we combine the advantage of stiff nanogels in RES-blockade with the superiority of soft nanogels in drug delivery leads to the optimum tumour inhibition effect, which is defined as mechano-boosting antitumour strategy. Clinical implications of stiffness-dependent RES-blockade are also confirmed by promoting antitumour efficacy of commercialized nanomedicines, such as Doxil and Abraxane.

Nanomedicine has been developed to promote drug delivery efficiency via the enhanced permeability and retention (EPR) effect for decades[1-3]. Doxil, Abraxane, and many other nanotherapeutics are commercialized and widely used for cancer therapy in clinical settings. Although pharmacokinetics, antitumour efficacy and safety of nanomedicines have achieved great benefits, only 0.7% of nanoparticles in average can be delivered to solid tumors[4]. Surface of nanoparticles is rapidly covered with complex serum proteins after i.v. injection, resulting in recognition and quick clearance from blood circulation by macrophages in reticuloendothelial system[5-8]. Modulating of nanoparticle size, surface charge, and targeting ligands has succeeded in decreasing liver clearance and prolonging blood circulation[9-12]. Among

these, PEGylation is a mature technology developed for stealthy drug delivery to impede liver clearance after surface modification by poly(ethylene glycol) (PEG)[13-15]. Although PEGylation renders nanoparticles stealthy and hard to be captured by macrophages[16-18], it causes another trouble that is restrained binding and limited internalization by tumor cells[19,20]. Because modification of nanoparticles leads similar uptake tendency for macrophages and tumor cells, contradiction always exists between liver clearance and tumor cellular internalization. Therefore, efficient strategies for decreasing blood clearance and increasing tumor accumulation are highly desirable.

Numerous strategies have been explored to inhibit the clearance function of RES, including depletion of macrophages by toxic

[1]National Engineering Research Center for Nanomedicine, College of Life Science and Technology, Huazhong University of Science and Technology, 430074 Wuhan, P. R. China. [2]Key Laboratory of Molecular Biophysics of Ministry of Education, College of Life Science and Technology, Huazhong University of Science and Technology, 430074 Wuhan, P. R. China. [3]Hubei Key Laboratory of Bioinorganic Chemistry and Materia Medical, Huazhong University of Science and Technology, 430074 Wuhan, P. R. China. [4]GBA Research Innovation Institute for Nanotechnology, 510530 Guangzhou, Guangdong, P. R. China. [5]Hubei Engineering Research Center for Biomaterials and Medical Protective Materials, Huazhong University of Science and Technology, 430074 Wuhan, P. R. China. ✉e-mail: zifuli@hust.edu.cn

molecules, obstacle of interaction between macrophages and nanoparticles, saturation of macrophages with large number of nanoparticles, or RES-blockade[12,14,21,22]. Toxic molecules such as dextran sulfate 500, methyl palmitate, gadolinium chloride and clodronate liposomes are used to kill and deplete macrophages to decrease the clearance of nanoparticles from blood circulation[21,23–25]. However, severe systemic toxicity comes along as well as unexpected bacterial infection may occur without the protection of macrophages. Safer methods are required. Mononuclear phagocyte system erythroblockade (MPS-erythroblockade), which is utilizing a low dose of allogeneic anti-erythrocyte antibodies and forcing MPS to clear erythrocyte, has been proven to increase the blood circulation half-life of nanoparticle formulations[26]. Pre-injecting cationized mannan-modified extracellular vesicles and then injecting drug-loaded nanocarriers fused with CD47-enriched exosomes, lead to prolonged circulation time and increased tumor accumulation[27]. Liposomes modified by CD47-derived peptide ligand can combine with macrophage membranes, inducing enduring enclose of membrane and silence of macrophages with lower dosage[22], while it is hard to predict whether potential immune suppression induced by CD47-derived peptide will cause unexpected tumor growth or metastasis[28–30]. Alternatively, pre-injection of large quantities of blank nanoparticles is demonstrated to be useful for temporarily and reversely saturation of macrophages to prolong blood circulation, and such treatment has been designated as RES-blockade strategy[12,31,32]. Clearance of large unilamellar liposomes could be blocked by small or large unilamellar liposomes and other particles, and mechanistic studies further revealed that opsonin was not the only factor and cellular saturation could also produce robust RES-blockade effect[33]. In 1960s, it was reported that administration of large dosage of either aggregated human serum albumin or gelatin could temporarily block RES and inhibit clearance of both similar or dissimilar particles, while a greater degree and longer duration were observed with similar particles. This RES-blockade effect was observed in both dog and man, leading to potential clinical applications[34]. Inevitably, a large quantity of nanoparticles in blood circulation may also bring additional burden in RES. Hence, improvement of RES-blockade efficiency is necessary.

To fully exploit RES-blockade effect, several factors have been investigated and optimized. It is found that 1.5 h is an appropriate time interval between RES-blockade and subsequent nanotherapeutic administration[12]. Besides, RES-blockade efficiency shows positive correlation with dosage and the number of 1 trillion is identified as threshold for inhibition of liver clearance[35]. Physicochemical properties of nanoparticles also play an essential role, for example, positive surface charge or large diameter leads to potent RES-blockade[12,31]. Mechanical properties as a widely-concerned parameter on nanomedicine design, has exerted a fundamental influence in drug delivery, especially in tumor penetration[36]. For example, soft DOX@3D-MPs and soft FA-PEG-modified silica nanocapsules can squeeze though gaps among tumor cells to achieve much deeper penetration than stiff counterparts, benefitting from excellent deformability to overcome hindrance coming from the dense extracellular matrix[37,38]. Nevertheless, the effect of nanoparticle mechanical properties, especially stiffness, on RES-blockade has not been investigated up to now. Furthermore, it is not clear how nanoparticles with distinctive mechanical properties affect macrophage biological behaviors.

Nanogel has been developed as a widely used nanocarrier owing to excellent stability, hydrophilicity, and biocompatibility[39]. Different monomers can provide multiple functions to make nanogels appropriate for multifarious applications[40–42]. Because of the unique network structure, nanogels exhibit flexible deformability and the mechanical properties are easily controlled by tuning crosslinking density[43–46]. Instead of traditional N-isopropylacrylamide (NIPAM) with lower volume phase transition temperature (VPTT) at around 32 °C and limited deformability at body temperature[47–49], we utilized

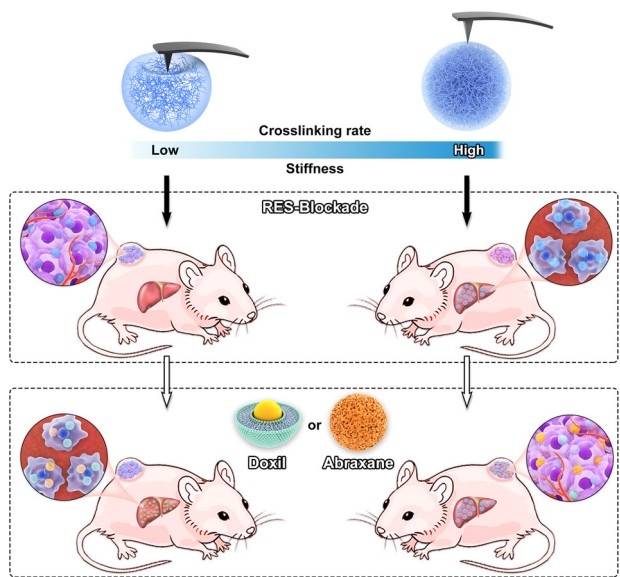

**Fig. 1 | Schematic illustration of mechano-boosting nanomedicine antitumour efficacy by blocking the reticuloendothelial system with stiff nanogels.** Nanogels with distinctive stiffness are injected preferentially and soft nanogels accumulate more in tumor, while stiff nanogels accumulate more in liver and temporarily block reticuloendothelial system (RES). Afterward, therapeutic nanoparticles such as soft DOX-loaded nanogels, commercialized nanomedicine Doxil and Abraxane are injected and antitumour efficacy is obviously promoted after RES-blockade by stiff nanogels.

N-isopropylmethacrylamide (NIPMAM) as monomer because of its higher VPTT to ensure extra inner space for deformation at 37 °C[50,51].

In this work, we prepared a series of poly(N-isopropylmethacrylamide-disulfide bond-methacrylic acid) (P(NIPMAM-ss-MAA)) nanogels with distinctive stiffness and found that taking full advantage of nanoparticles mechanical properties by blocking RES with stiff nanogels and delivering therapeutic DOX with soft nanogels could achieve the optimum antitumour efficacy. After labeled by Rhodamine B (RhB) or indocyanine green (ICG), nanogels could be used for both in vitro and in vivo imaging. Meanwhile, DOX-loading ability conferred nanogels on cancer therapy. DOX-loaded soft nanogels presented excellent deformability and led to high tumor accumulation as well as potent tumor inhibition. Nevertheless, their antitumour efficacy was still limited by RES clearance and could be enhanced by pre-injection of stiff nanogels via suppressing internalization capacity of macrophages. Mechanistical studies revealed that stiff nanogels blocked RES by suppressing clathrin-mediated endocytosis. Importantly, antitumour efficacy of commercialized nanomedicines, such as Doxil and Abraxane, could also be boosted with stiff nanogels but not soft nanogels, demonstrating the clinical implications of stiffness-dependent RES-blockade strategy (Fig. 1).

## Results
### Synthesis and characterization of P(NIPMAM-ss-MAA) nanogels with distinctive mechanical properties
To investigate how the mechanical properties affected antitumour efficacy of nanomedicine, we prepared a series of P(NIPMAM-ss-MAA) nanogels with distinctive stiffness to load DOX. P(NIPMAM-ss-MAA) nanogels were synthesized via emulsion polymerization. Temperature-responsive monomer N-isopropylmethacrylamide (NIPMAM) was used to construct nanogels with outstanding deformability at physiological temperature. pH-responsive monomer methacrylic acid (MAA) was introduced to enhance the hydrophilicity of nanogels for stability and biocompatibility under physiological conditions, as well as provide negative surface charge for drug loading via electrostatic interaction.

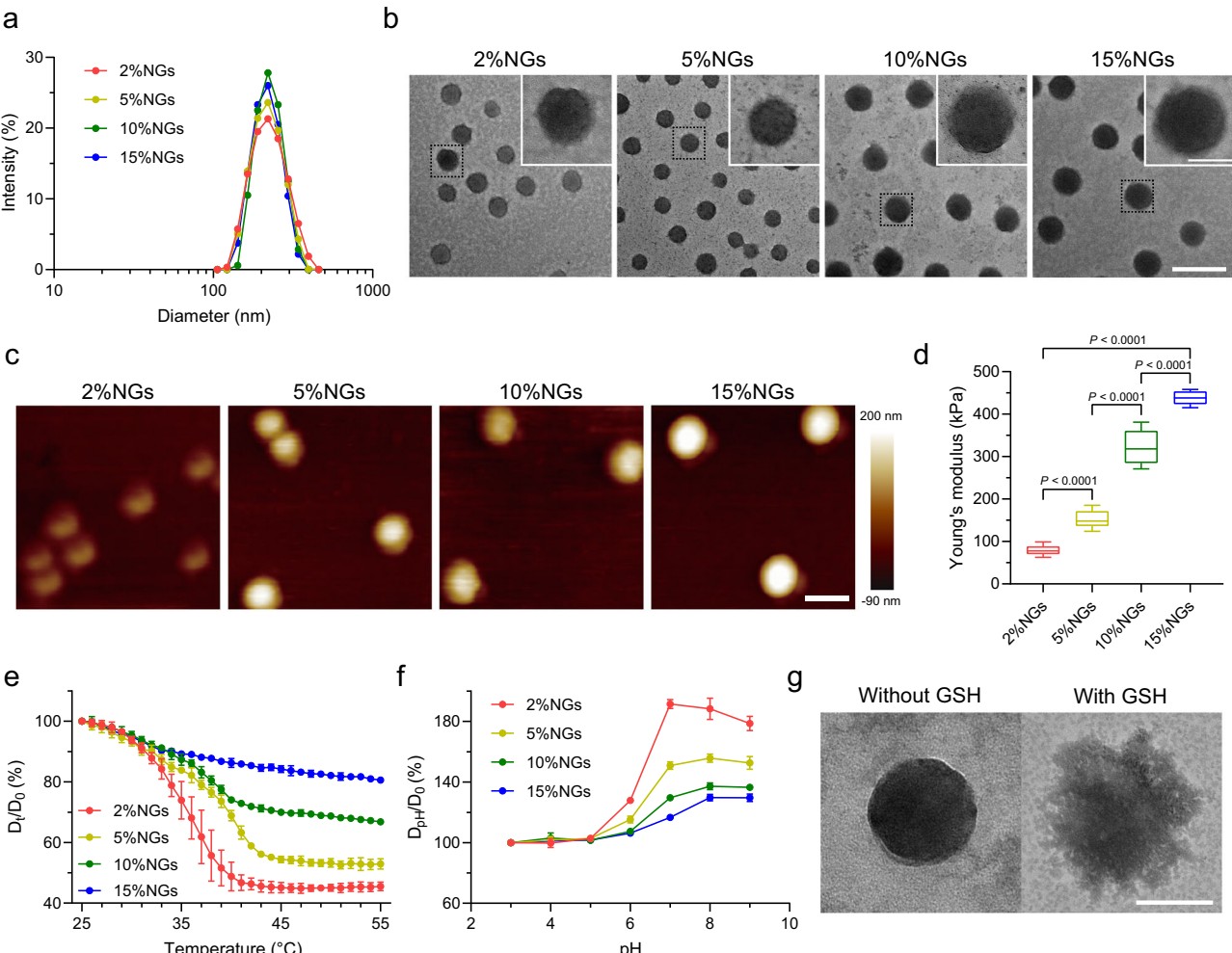

**Fig. 2 | Characterization of nanogels with distinctive stiffness. a** Size distribution of nanogels with distinctive stiffness in PBS at 37 °C. **b** TEM images of nanogels with distinctive stiffness. Scale bar = 500 nm in low magnification and Scale bar = 100 nm in high magnification. **c** AFM images of nanogels with distinctive stiffness in liquid phase. Scale bar = 200 nm. **d** Young's modulus of nanogels with distinctive stiffness in liquid phase. Box plots indicate median (middle line), 25th, 75th percentile (box) and minimum and maximum (whiskers) ($n = 15$ independent replicates). **e** Temperature-responsiveness of nanogels with distinctive stiffness in $H_2O$. Data are presented as mean values ± SD ($n = 3$ independent replicates). **f** pH-responsiveness of nanogels with distinctive stiffness in $H_2O$. Data are presented as mean values ± SD ($n = 3$ independent replicates). **g** TEM images of 2%NGs incubated with or without GSH for 24 h. Scale bar = 100 nm. Statistical significance was calculated by one-way ANOVA. Source data are provided as a Source Data file.

Crosslinker N,N′-bis(acryloyl)cystamine (BAC) was glutathione-responsive (GSH-responsive) to bestow nanogels with biodegradability. Furthermore, the molar ratio between crosslinker BAC and monomer NIPMAM was regulated for the preparation of nanogels with distinctive mechanical properties. Nanogels with cross-linking rate of 2%, 5%, 10%, and 15% are designated as 2%NGs, 5%NGs, 10%NGs, and 15% NGs, respectively. To exclude the influence of nanogel size, the amount of surfactant sodium dodecyl sulfate (SDS) was adjusted to ensure similar size distribution of nanogels with varied mechanical properties in PBS at 37 °C.

A similar hydrodynamic diameter of around 220 nm in PBS at 37 °C of nanogels was confirmed by dynamic light scattering (DLS) examination (Fig. 2a). The nanogels showed similar negative surface charge ranging from −13.5 ± 0.2 mV to −23.8 ± 0.4 mV in $H_2O$ (Supplementary Fig. 1a). Transmission electron microscopy (TEM) and atomic force microscopy (AFM) images revealed that nanogels were monodispersed with spherical morphologies (Fig. 2b, c). A slight difference in diameter among TEM images resulted from shrinkage of nanogel network structure after drying during the preparation of TEM samples. Young's modulus of nanogels were measured in liquid phase, which was considered as the stiffness of nanogels. Young's modulus of

nanogels was positively associated with cross-linking rate of nanogels, of which the softest 2%NGs was 79.0 ± 9.9 kPa, 5%NGs 151.5 ± 18.9 kPa, 10%NGs 321.2 ± 36.3 kPa and the stiffest 15%NGs 439.2 ± 14.9 kPa (Fig. 2d). Because of the monomers NIPMAM and MAA, temperature-responsiveness and pH-responsiveness of the nanogels were evaluated. The nanogels shrank with temperature increase relying on the interruption of hydrogen bonds between water and polymer segments[47,48], or swelled with pH increase because of the deprotonation of carboxyl groups in nanogels[52,53] (Fig. 2e, f). Absolute values of temperature-responsive and pH-responsive diameter variation curve were also presented (Supplementary Fig. 1b, c). Take 2%NGs as an example, the volume of the nanogels shrinks by 10.62 times as the temperature increases from 25 °C to 55 °C while the volume swells by 5.70 times by raising pH from 3 to 9. For 15%NGs, the volume only decreases by 1.92 times with increased temperature and expands by 2.18 times with increased pH values. These results corroborated that soft nanogels showed better deformability compared to stiff ones, while the softest 2%NGs was the most deformable and the stiffest 15% NGs was the least deformable. GSH-responsiveness was characterized by TEM with or without incubation in 10 mmol/L GSH solution. After incubation with GSH, the structure of 2%NGs was destructed (Fig. 2g).

Hydrophilic DOX·HCl as an antitumour drug was loaded onto nanogels via electrostatic interaction to obtain DOX@2%NGs, DOX@5%NGs, DOX@10%NGs and DOX@15%NGs. Herein, 2%NGs and DOX@2%NGs, 15%NGs and DOX@15%NGs were taken into comparison to confirm that loading DOX exerted insignificant influence on the properties of nanogels (Supplementary Fig. 2). Diameter distribution in PBS at 37 °C and temperature-responsiveness of nanogels maintained consistency after loading DOX (Supplementary Fig. 2a, b). TEM and AFM images showed no variation on size and morphology (Supplementary Fig. 2c, d). Importantly, no significant difference in Young's modulus after loading DOX could be detected, $75.9 \pm 10.2$ kPa for 2% NGs and $71.4 \pm 10.2$ kPa for DOX@2%NGs, $421.7 \pm 12.2$ kPa for 15%NGs and $421.2 \pm 16.7$ kPa for DOX@15%NGs, respectively (Supplementary Fig. 2e). Furthermore, stability of DOX@2%NGs and DOX@15%NGs was evaluated in PBS and FBS at 37 °C. Both nanogels were quite stable after incubation for 4 days (Supplementary Fig. 2f). Collectively, a series of nanogels with similar properties except for stiffness have been prepared and loaded with DOX to investigate the mechanical-dependent drug delivery behaviors.

## Advantages of soft nanogels in tumor targeting drug delivery

Although many physical and biochemical properties of nanoparticles have been optimized, such as size, surface charge, morphology, structure, components, and targeting ligands, to improve antitumour efficacy of nanomedicine, complexed tumor mechanical microenvironment was still an obstacle to hinder penetration of nanomedicine[1,54]. Previous studies have demonstrated that mechanical properties, especially stiffness of nanomedicine profoundly affected five critical processes of tumor targeting delivery[36]. As for tumor cells, some nanoparticle systems showed higher cellular uptake efficiency in their soft counterparts than stiff ones[37,38]. Herein, we focused on the stiffness-dependent effects on 4T1 murine breast cancer cells and tumors. Cytotoxicity of nanogels was evaluated in 4T1 cells after incubation for 24, 48, and 72 h, no cytotoxicity was observed (Fig. 3a and Supplementary Fig. 3). To avoid the influence of DOX on 4T1 cells including time-dependent drug release and cytotoxicity, Rhodamine-labeled nanogels via stable covalent linkage with different stiffness were prepared for tracking and quantification. After incubation with Rhodamine-labeled nanogels for 4 h, 4T1 cells were examined by flow cytometry. The fluorescent intensity of cells was measured (Supplementary Fig. 4) and relative cellular uptake was calculated according to the relative fluorescent intensity of different Rhodamine-labeled nanogels (Supplementary Table 1), leading to a conclusion that soft nanogels (2%NGs and 5%NGs) showed higher cellular uptake efficiency than stiff counterparts (10%NGs and 15%NGs) (Fig. 3b). Afterwards, stiffness-dependent cytotoxicity was evaluated by MTT assay with DOX-loaded nanogels, the result confirmed that soft 2%NGs were more efficient in killing 4T1 cancer cells compared to stiff 15%NGs (Fig. 3c). The $IC_{50}$ was 3.1 μg/mL and 7.1 μg/mL of DOX for 2%NGs and 15%NGs, respectively.

In addition, soft nanoparticles exhibited excellent deformability to squeeze through dense tumor matrix, achieving deeper penetration and higher tumor accumulation[37,38]. To investigate the permeability of nanogels with different stiffness, Matrigel was utilized to simulate the extracellular matrix ex vivo. Rhodamine-labeled 2%NGs and 15%NGs were incubated above Matrigel for 6 h and removed, deeper penetration was observed on 2%NGs as compared to 15%NGs (Fig. 3d). Similarly, in three-dimensional (3D) stroma-rich tumor spheroids, 2%NGs could penetrate to the core area of spheroids after incubation for 6 h, whereas 15%NGs could only penetrate for 10 μm of depth (Fig. 3e). To further investigate whether outstanding permeability of soft nanogels was beneficial for tumor accumulation, we labeled DOX-loaded nanogels with ICG via π−π interaction for in vivo imaging[55,56]. The mice were sacrificed and tumors were harvested at 1, 4, 8, and 24 h post injection of ICG-loaded nanogels with different stiffness, then the tumors were

imaged by in vivo imaging system (IVIS) and the fluorescent intensity was quantified. The result demonstrated that 2%NGs could achieve higher accumulation at tumor site than 15%NGs (Fig. 3f−g) and accumulation of nanogels in major organs was presented in Supplementary Fig. 5. Meanwhile, in vivo imaging was captured at 1, 2, 4, 8, 12, and 24 h post injection and consistent result was observed (Supplementary Fig. 6). In consideration of influence coming from the release of ICG, Rhodamine-labeled nanogels were also used to investigate the accumulation of nanogels. However, short fluorescent wavelength ($\lambda_{ex} = 568$ nm, $\lambda_{em} = 583$ nm) of Rhodamine B led to dissatisfactory in vivo imaging effect, the mice were sacrificed and tumors were harvested for ex vivo imaging 4 h post injection (Supplementary Fig. 7a). The semi-quantification of fluorescent intensity gave the same conclusion that higher amount of soft 2%NGs accumulated at tumor site than stiff 15%NGs (Supplementary Fig. 7b).

Furthermore, higher tumor accumulation and deeper penetration led to better antitumour efficacy[37,57]. To evaluate stiffness-dependent antitumour efficacy, 4T1 subcutaneous tumor model was established and divided into five groups including control (G1), free DOX (G2), DOX@2%NGs (G3), DOX@10%NGs (G4), DOX@15%NGs (G5). At the end of the experiment, Fig. 3h revealed that free DOX could only achieve feeble antitumour efficacy. Although DOX@15% NGs could accumulate at tumor site via EPR effect, it showed similar antitumour efficacy as free DOX because of limited permeability with tumor volume of around $493 \pm 110$ mm$^3$. Better antitumour efficacy was obtained by DOX@10%NGs with tumor volume of around $394 \pm 67$ mm$^3$ and the best was by DOX@2%NGs with around $297 \pm 78$ mm$^3$ relying on better permeability ascribed to excellent deformability of soft nanogels (Fig. 3j). The tumors were harvested after the experiment and weighted for further proof of stiffness-dependent antitumour efficacy. Similar result was obtained to tumor volume (Fig. 3i). Meanwhile, volume- and weight-based tumor inhibition rates were calculated, revealing more than 4.5-fold higher antitumour efficacy from the soft DOX@2%NGs than that from the stiff DOX@15%NGs (Supplementary Fig. 8). These results suggested that antitumour efficacy was negatively associated with stiffness of nanogels. The harvested tumors were fixed with 4% paraformaldehyde for H&E, Tunel and Ki67 staining to evaluate necrosis, apoptosis, and proliferation of tumor cells. After stained by H&E, sparser distribution of cells could be observed in tumors treated by softer nanogels. The percentage of the area of apoptotic cells increased, while proliferative cells decreased, with the decrease of stiffness of nanogels (Supplementary Fig. 9). Besides, enhancement of tumor penetration benefitting from excellent deformability of soft nanogels was evaluated by fluorescent imaging of CD31 antibody-labeled blood vessels and DOX-loaded nanogels. Supplementary Fig. 10 demonstrated that the distance of soft DOX@2%NGs away from blood vessels was around $104 \pm 24$ μm, which was much longer than $44 \pm 18$ μm of stiff DOX@15%NGs. To investigate whether nanogels of varied stiffness could lead to unexpected side effects, body weight was measured, major organs were harvested for H&E staining and blood was collected for blood biochemical analysis and blood routine examine. The body weight showed temporary decrease in G3 and G4, which might be caused by higher accumulation in major organs, and recovered over time after drug administration (Fig. 3k). Meanwhile, H&E staining exhibited no obvious toxicity in major organs after treatment (Supplementary Fig. 11). In blood biochemical analysis and blood routine examine, the variation of white blood cells (WBC) could be observed and the number of WBC were always positively corelated with tumor volume, which could be a result of tumor malignancy and inflammation[58,59], while all the physiological indicators were still in the normal range (Supplementary Fig. 12). Taken together, DOX-loaded soft nanogels exhibited superior antitumour efficacy benefitting from higher cellular uptake and deeper tumor penetration than stiff nanogels.

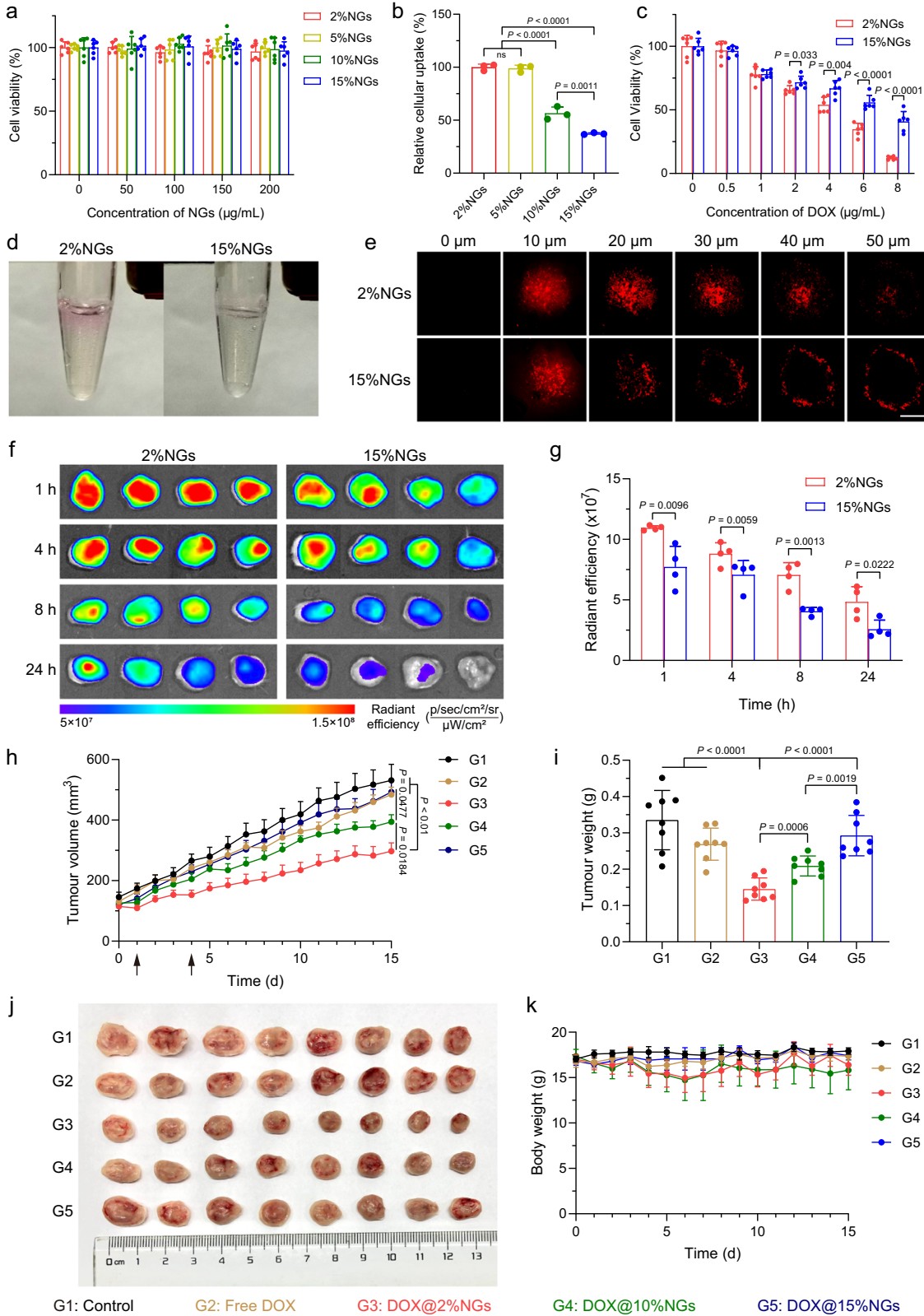

G1: Control          G2: Free DOX          G3: DOX@2%NGs          G4: DOX@10%NGs          G5: DOX@15%NGs

## Advantages of stiff nanogels in RES-blockade

Although soft nanogels could achieve excellent tumor penetration and accumulation (Fig. 3), they were facing a non-negligible problem that was rapid clearance from blood circulation after i.v. injection by reticuloendothelial system[6,60]. RES-blockade strategies were developed to overcome this issue by reducing unexpected clearance and improving nanoparticle tumor accumulation[25,61]. Next, we sought to assess the

influence of nanogels stiffness on RES-blockade. At the end of in vivo imaging of stiffness-dependent tumor accumulation, the mice were sacrificed, tumors and major organs were collected for ex vivo imaging (Fig. 4a). The fluorescent intensity was semi-quantified and relative accumulation in tumor to liver was calculated (Fig. 4b and Supplementary Fig. 13). Although ICG-loaded 2%NGs showed excellent tumor accumulation, we noticed that ICG-loaded 15%NGs retained more in

**Fig. 3 | Advantages of soft nanogels in tumor targeting drug delivery.**
**a** Cytotoxicity of nanogels with different stiffness for 24 h. Data are presented as mean values ± SD ($n = 6$ biological independent replicates). **b** Relative cellular uptake efficiency of Rhodamine-labeled nanogels with different stiffness in 4T1 cells. Data are presented as mean values ± SD ($n = 3$ biological independent replicates). **c** Stiffness-dependent cytotoxicity of DOX-loaded nanogels with different stiffness in 4T1 cells. Data are presented as mean values ± SD ($n = 6$ biological independent replicates). **d** Permeability of Rhodamine-labeled nanogels with different stiffness in Matrigel. **e** Permeability of Rhodamine-labeled nanogels with different stiffness in 3D tumor spheroids. Scale bar = 200 μm. **f** Ex vivo fluorescent images of tumor accumulation of ICG-loaded nanogels with different stiffness. **g** Semi-quantification of the quantity of ICG-loaded nanogels with different

stiffness in tumor. Data are presented as mean values ± SD ($n = 4$ biological independent replicates). **h** Tumor growth profiles of 4T1 tumor-bearing mice treated by DOX-loaded nanogels with different stiffness at DOX dosage of 4 mg/kg. Data are presented as mean values ± SEM ($n = 8$ biological independent replicates). **i** Weight of tumors after treatment. Data are presented as mean values ± SD ($n = 8$ biological independent replicates). **j** Photograph of tumors extracted from the mice after treatment. **k** Body weight profiles of mice after treatment. Data are presented as mean values ± SD ($n = 8$ biological independent replicates). Statistical significance of **b** was calculated by one-way ANOVA. Statistical significance of (**c, g, h, i**) was calculated by unpaired two-sided Student's *t* test. G1, Control; G2, Free DOX; G3, DOX@2%NGs; G4, DOX@10%NGs; G5, DOX@15%NGs. Source data are provided as a Source Data file.

liver than 2%NGs. These results suggested stiff nanogels might lead to higher RES-blockade efficiency by inhibiting clearance function of liver for a longer period than soft counterparts. To this end, time interval between RES-blockade and drug administration needed to be determined first. We performed in vitro cellular uptake experiment and found that the RAW 264.7 cells were saturated at around 1.5 h after incubation with Rhodamine-labeled 15%NGs (Supplementary Fig. 14). However, in vivo imaging of ICG-loaded 2%NGs with different time interval from 0.5 to 3 h presented no obvious differences, both in 2%-blockade and 15%-blockade groups (Supplementary Fig. 15). In consideration of saturating macrophages and no more increase of tumor accumulation with prolonged time interval, 1.5 h was set as the time interval between RES-blockade and subsequent drug administration, in consistence with previous work[12]. Accordingly, we investigated RES-blockade efficiency of nanogels with distinctive stiffness by detecting fluorescent intensity of ICG-loaded 2%NGs in tumor and liver after 1.5 h of pre-blockade by 2%NGs or 15%NGs at dosage of 200 mg/kg. The mice were sacrificed and tumors were harvested at 1, 4, 8, and 24 h post injection of ICG-loaded 2%NGs, then the tumors were imaged by IVIS and the fluorescent intensity was quantified. Figure 4c–f revealed that RES-blockade by soft 2%NGs hardly inhibit liver clearance nor promote tumor accumulation of the subsequent ICG-loaded 2%NGs, whereas RES-blockade by stiff 15%NGs conspicuously enhanced tumor accumulation and suppressed liver clearance. Meanwhile, the mice were imaged at 1, 2, 4, 8, 12, and 24 h post injection of ICG-loaded 2%NGs to continuously observe the variation of fluorescent intensity in tumor and liver to further confirm high RES-blockade efficacy of 15%NGs (Supplementary Fig. 16). In addition to enhanced tumor accumulation, influence on blood clearance of nanomedicine was another important parameter of RES-blockade. Similarly, 2%-blockade induced negligible improvement in the pharmacokinetics of ICG-loaded 2%NGs. In contrast, RES-blockade by 15%NGs significantly prolonged blood half-life time of ICG-loaded 2%NGs from 0.70 ± 0.14 h to 2.29 ± 0.36 h, increased AUC from 8.90 ± 2.15 mg L$^{-1}$ h$^{-1}$ to 41.49 ± 8.71 mg L$^{-1}$ h$^{-1}$, and decreased plasma clearance from 0.65 ± 0.14 L kg$^{-1}$ h$^{-1}$ to 0.14 ± 0.03 L kg$^{-1}$ h$^{-1}$, as a result of inhibited liver clearance (Supplementary Fig. 17 and Supplementary Table 2). Stiffness-dependent RES-blockade efficiency was also evaluated on RAW 264.7 cells in vitro and the operation procedure was illustrated in Fig. 4g. Consistently, higher blockade efficiency was observed in RAW 264.7 cells by stiff 15%NGs, whereas no evident inhibition effect was observed in RAW 264.7 cells blocked by soft 2%NGs (Fig. 4h and Supplementary Fig. 18). These results collectively illustrated that stiff nanogels were more efficient in blocking RES than soft counterparts, contributing to promotion on tumor accumulation and inhibition of liver clearance of subsequent administrated nanogels.

## Stiffness as a key parameter in RES-blockade

Numerous parameters have been investigated and proved to be closely associated with efficiencies of RES-blockade, such as surface charge, diameter, and dosage of nanoparticles[12,31,61]. Dosage has been commonly regarded and utilized as a key parameter in RES-blockade strategies by many nanoparticles[12,62]. A dosage of 1 trillion

nanoparticles per mouse has recently been considered as threshold for inhibited macrophages uptake, decreased liver clearance, prolonged blood circulation, and enhanced tumor accumulation[35]. For this reason, we sought to investigate whether stiff 15%NGs showed dosage-dependent RES-blockade efficiency and possessed a threshold. Dosages of 15%NGs including 100 mg/kg, 200 mg/kg were used to block RES for 1.5 h, then ICG-loaded 2%NGs were administered to detect fluorescent intensity in tumor and liver. The tumors were harvested and ex vivo imaged at 1, 4, 8 and 24 h post injection of ICG-loaded 2%NGs and the fluorescent intensity was quantified. Figure 5a–d shows that the dosage of 100 mg/kg could inhibit liver clearance while tumor accumulation was not improved. In comparison, tumor accumulation of mice blocked by dosages of 200 mg/kg was significantly promoted and liver clearance was inhibited. In vivo imaging also confirmed that dosage of 200 mg/kg exhibited more excellent RES-blockade efficacy compared to 100 mg/kg, while elevation of dosage to 300 mg/kg could not further enhance tumor accumulation (Supplementary Fig. 19). In consideration of biocompatibility and safety, dosage of 300 mg/kg was abandoned because of the potential burden on liver and 200 mg/kg was selected as appropriate dosage for RES-blockade. For subsequent RES-blockade studies, the dosage of 200 mg/kg is utilized unless otherwise specified. Dosage-dependent RES-blockade efficiency was further evaluated in RAW 264.7 cells in vitro and the operation procedure was illustrated in Supplementary Fig. 20. The result demonstrated that the cellular uptake efficiency was gradually inhibited with the increase of 15%NGs dosage, suggesting RES-blockade efficiency was positively correlated with the dosage of nanoparticles.

To confirm whether the number of nanogels used for RES-blockade also abided by the threshold of 1 trillion per mouse, we quantified the number of nanogels with varied stiffness used for RES-blockade. First, we analyzed the number concentration of nanogels by nanoparticle tracking analysis (NTA). Consistent with the results of DLS (Fig. 2a), the concentration-dependent diameter distribution demonstrated that nanogels with distinctive stiffness were monodispersed with similar diameter (Supplementary Fig. 21). According to the tests from NTA, soft 2%NGs possessed the largest number of nanogel particles as (3.78 ± 0.76) × 10$^{11}$ particles per milligram, whereas the stiff 15%NGs possessed the smallest as (4.80 ± 0.37) × 10$^{10}$ particles per milligram (Supplementary Fig. 22). As designed, softer nanogels contained larger quantity of particles than stiff counterparts due to the lower cross-linking rate and density. By calculating the number of nanogels injected into the mice, we found that the number of 2%NGs for RES-blockade was (1.51 ± 0.30) × 10$^{12}$ per mouse while 15%NGs was (1.91 ± 0.15) × 10$^{11}$ per mouse (Fig. 5e). In comparison with 1 trillion nanoparticles per mouse, 1.5-fold larger quantity of 2%NGs was used with negligible RES-blockade efficiency, while 15%NGs could achieve excellent RES-blockade efficiency with only 20% of quantity. These results suggest that dosage is an important parameter but not the only one, mechanical properties especially stiffness of blockaded nanoparticles might be a significant parameter for RES-blockade.

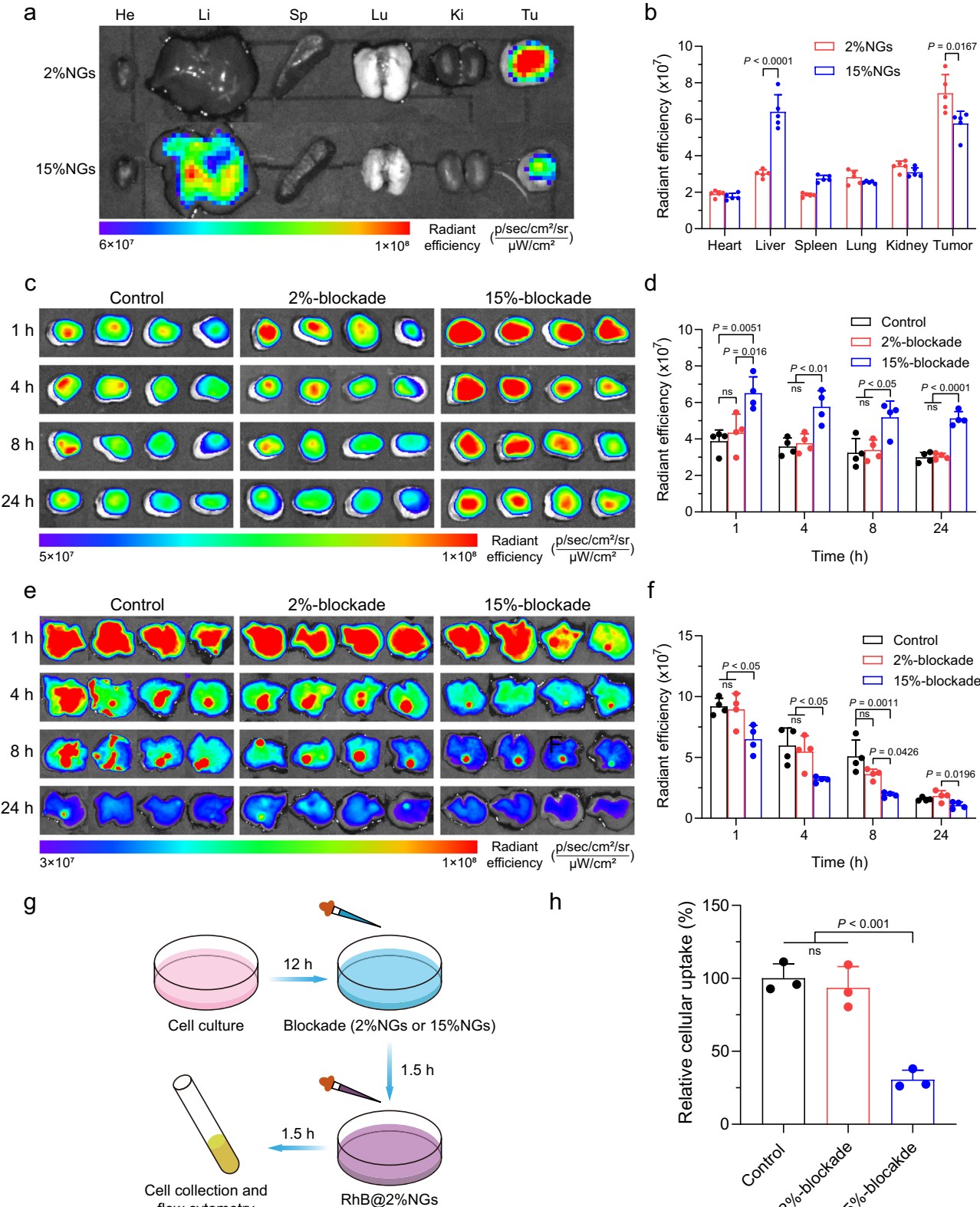

**Fig. 4 | Advantages of stiff nanogels in RES-blockade. a** Ex vivo fluorescent images of biodistribution of ICG-loaded nanogels with different stiffness at 24 h after injection. He heart, Li liver, Sp spleen, Lu lung, Ki kidney, Tu tumor. **b** Semi-quantification of the quantity of ICG-loaded nanogels with different stiffness in tumor and major organs ex vivo. Data are presented as mean values ± SD ($n = 5$ biological independent replicates). Ex vivo fluorescent images of (**c**) tumor accumulation and (**e**) liver accumulation of ICG-loaded 2%NGs after RES-blockade by nanogels with different stiffness at dosage of 200 mg/kg. Semi-quantification of

the quantity of ICG-loaded 2%NGs in (**d**) tumor and (**f**) liver. Data are presented as mean values ± SD ($n = 4$ biological independent replicates). **g** Illustration of in vitro stiffness-dependent RES-blockade experiment in RAW 264.7 cells. **h** Different blockade efficiency of nanogels with different stiffness in RAW 264.7 cells at concentration of 400 µg/mL. Data are presented as mean values ± SD ($n = 3$ biological independent replicates). Statistical significance of (**d, f, h**) was calculated by one-way ANOVA. Statistical significance of (**b**) was calculated by unpaired two-sided Student's $t$ test. Source data are provided as a Source Data file.

To further understand how nanogel dosage and stiffness affected RES-blockade efficiency, we investigated nanogels cellular uptake with RAW 264.7 cells and revealed the impact of nanogels with distinctive stiffness on RAW 264.7 cells. First, stiffness-dependent cellular uptake was evaluated. Supplementary Fig. 23c demonstrated that soft 2%NGs were internalized at around threefold higher efficiency than stiff 15% NGs. Exocytosis of nanogels also proceeded at the same time and Supplementary Fig. 23f showed that similar amount of nanogels was excreted and around 40% of both 2%NGs and 15%NGs were detected after 1.5 h of exocytosis. These results demonstrated that smaller quantity of stiff 15%NGs remained within RAW 264.7 cells and suggested that the stiffness but not the quantity of nanogels played a key role in mediating subsequent cellular uptake efficiency of RAW 264.7 cells. Considering the decrease of cellular uptake did not rely on saturation of macrophages, we speculated that this might result from incapability of internalization. For this reason, we sought to study whether the mRNA transcription of endocytosis-related protein was regulated post-treatment with nanogels of distinctive stiffness. Among endocytosis pathways, clathrin-mediated endocytosis was an important pathway for macrophages uptaking hybrid nanoparticles with around 200 nm and nanogels with 211 kPa[43,63]. Suppression of clathrin-mediated endocytosis might be the reason why 15%NGs could induce higher RES-blockade efficiency. Figure 5f showed that transcription of clathrin-related subunits, including Clta and Cltc, were significantly downregulated with the increase of nanogel stiffness. To confirm macrophages internalized nanogels via clathrin-mediated endocytosis, clathrin inhibitor Pitstop 2 was used to treat macrophages. Figure 5g demonstrated that uptake efficiency of nanogels decreased with increased concentration of Pitstop 2, suggesting nanogels were internalized by RAW 264.7 cells via clathrin-mediated endocytosis and inhibition of this pathway could obviously decrease internalization of nanogels. It has been reported that the heterotetrameric adapter protein complex-2 (AP-2) was an essential component of clathrin-coated vesicles by linking clathrin lattice with underlying membrane[64,65]. So, transcription of AP-2-related subunits were also investigated and Supplementary Fig. 24 shows that transcription of subunits Ap2a1, Ap2a2, and Ap2b1 were also conspicuously downregulated after incubation with stiff 15%NGs. Together, clathrin-mediated cellular uptake of RAW 264.7 cells was interrupted after incubation with 15%NGs, resulting in lower uptake efficiency of subsequent nanoparticles. This might be the reason why stiff 15%NGs could achieve higher RES-blockade efficiency than 2%NGs. Meanwhile, this also provided an explanation why stiff nanogels showed lower cellular uptake efficiency than soft nanogels.

Despite most of nanoparticles were cleared by RES, hepatobiliary was another way for liver clearance of nanoparticles[6]. Soft nanogels could pass through liver sinusoidal fenestration more easily relying on better deformability and directly interact with hepatocytes, leading to decreased internalization by macrophages[66]. In addition, loose structure of soft nanogels resulted in faster biodegradation and further limited RES-blockade efficiency. Accordingly, we investigated the retention of nanogels in liver. ICG-loaded 2%NGs and 15%NGs were injected into the mice, the mice were sacrificed and the livers were harvested for ex vivo imaging at 1, 4, 8, and 24 h post injection. We found that the fluorescent intensity of stiff 15%NGs was continuously higher than soft 2%NGs (Fig. 5h). After 24 h post injection, around 25% of the fluorescent intensity of 15%NGs remained and kept sustained since 8 h, while the fluorescent intensity of 2%NGs continuously decreased to around 13% (Fig. 5i). Being crosslinked at low density, 2% NGs could be easily degraded and eliminated from liver. In contrast, 15%NGs was stiff enough to withstand biodegradation and elimination by liver and inhibited internalization of macrophages for a longer time, leading to long-lasting RES-blockade. Collectively, these results suggest that, besides the dosage of nanoparticles, stiffness is also a critical parameter for RES-blockade.

## Stiff-blockade combined with soft delivery contributing to enhanced antitumour efficacy

It has been demonstrated that soft nanogels could achieve better antitumour efficacy (Fig. 3) while stiff nanogels could achieve better RES-blockade efficiency (Fig. 4). We postulated that the combination of stiff-blockade with soft delivery would contribute to augmented antitumour efficacy, if nanogels with varied stiffness could take full advantages at different stages of drug delivery. To that end, 4T1 subcutaneous tumor model was established and divided into seven groups including control (G1), free DOX (G2), DOX@2%NGs (G3), DOX@15%NGs (G4), 2%-blockade + DOX@2%NGs (G5), 15%-blockade + DOX@2%NGs (G6) and 15%-blockade + DOX@15%NGs (G7). The drug administration strategy is illustrated in Fig. 6a. Consistent with Fig. 3, neither free DOX (G2) nor DOX@15%NGs (G4) showed obvious tumor suppression effect because of the lack of tumor targeting ability or poor permeability (Fig. 6b). DOX@2%NGs (G3) exhibited much better antitumour efficacy relying on tumor targeting ability and excellent permeability. 2%-blockade + DOX@2%NGs (G5) showed limited decrease of tumor volume compared to DOX@2%NGs (G3) because of no obvious improvement in tumor accumulation and noneffective RES-blockade (Fig. 4). In striking contrast, 15%-blockade remarkably promoted the effect of DOX@2%NGs (G6) relying on effective RES-blockade and higher tumor accumulation, further decreasing around 27% of tumor volume than G3. G6 achieved the best antitumour efficacy among all groups with the lowest tumor volume of $380 \pm 43$ mm$^3$. Similarly, 15%-blockade significantly promoted antitumour effect of DOX@15%NGs (G7) by around 20% as compared with G4 (Fig. 6b, d). These results corroborated that RES-blockade with stiff nanogels boosted antitumour efficacy of DOX-loaded nanogels and that the combination of stiff-blockade with soft-delivery by taking full advantages of nanogels with varied stiffness at different stages achieves the optimum antitumour strategy. The tumors were harvested after the experiment and weighed. Consistent result was obtained (Fig. 6c). Meanwhile, volume- and weight-based tumor inhibition rate was calculated (Supplementary Fig. 25). Consistently, more than twofold higher tumor inhibition rate was observed in DOX@2% NGs (G3) than DOX@15%NGs (G4). 2%-blockade (G5) showed no obvious promotion on DOX@2%NGs (G3). 15%-blockade (G6 and G7) promoted more than 1.2-fold tumor inhibition rate of both DOX@2% NGs (G3) and DOX@15%NGs (G4). Also, G6 presented the best antitumour efficacy with tumor inhibition rate of around 50%, ascribing to full advantages of nanogel mechanical properties. The harvested tumors were fixed with 4% paraformaldehyde for H&E, Tunel and Ki67 staining to evaluate necrosis, apoptosis, and proliferation of tumor cells (Supplementary Fig. 26). After stained by H&E, the sparsest distribution of cells could be observed in tumors treated by 15%-blockade + DOX@2%NGs among all groups. Likewise, the highest percentage of apoptotic cells and the lowest percentage of proliferative cells were detected in G6 than all other treatments (Fig. 6f, g).

To investigate whether RES-blockade led to unexpected side effects, body weight was measured, major organs were harvested for H&E staining, and blood was collected for blood biochemical analysis and blood routine examine. Slight loss of body weight could be observed during treatment which might be caused by higher accumulation of DOX-loaded nanogels in organs and tissues, however, the body weight recovered to normal level after treatment and negligible toxicity toward major organs could be observed in H&E staining at the end of experiment (Fig. 6e and Supplementary Fig. 27). In blood biochemical analysis and blood routine examine, in addition of the varied WBC level, the ALT level obviously increased in G7 might result from burden on liver biodegrading and clearing 15%NGs. Meanwhile, all the physiological indicators were still in the normal range (Supplementary Fig. 28). These results indicated the combination of stiff-blockade and soft delivery augmented antitumour efficacy.

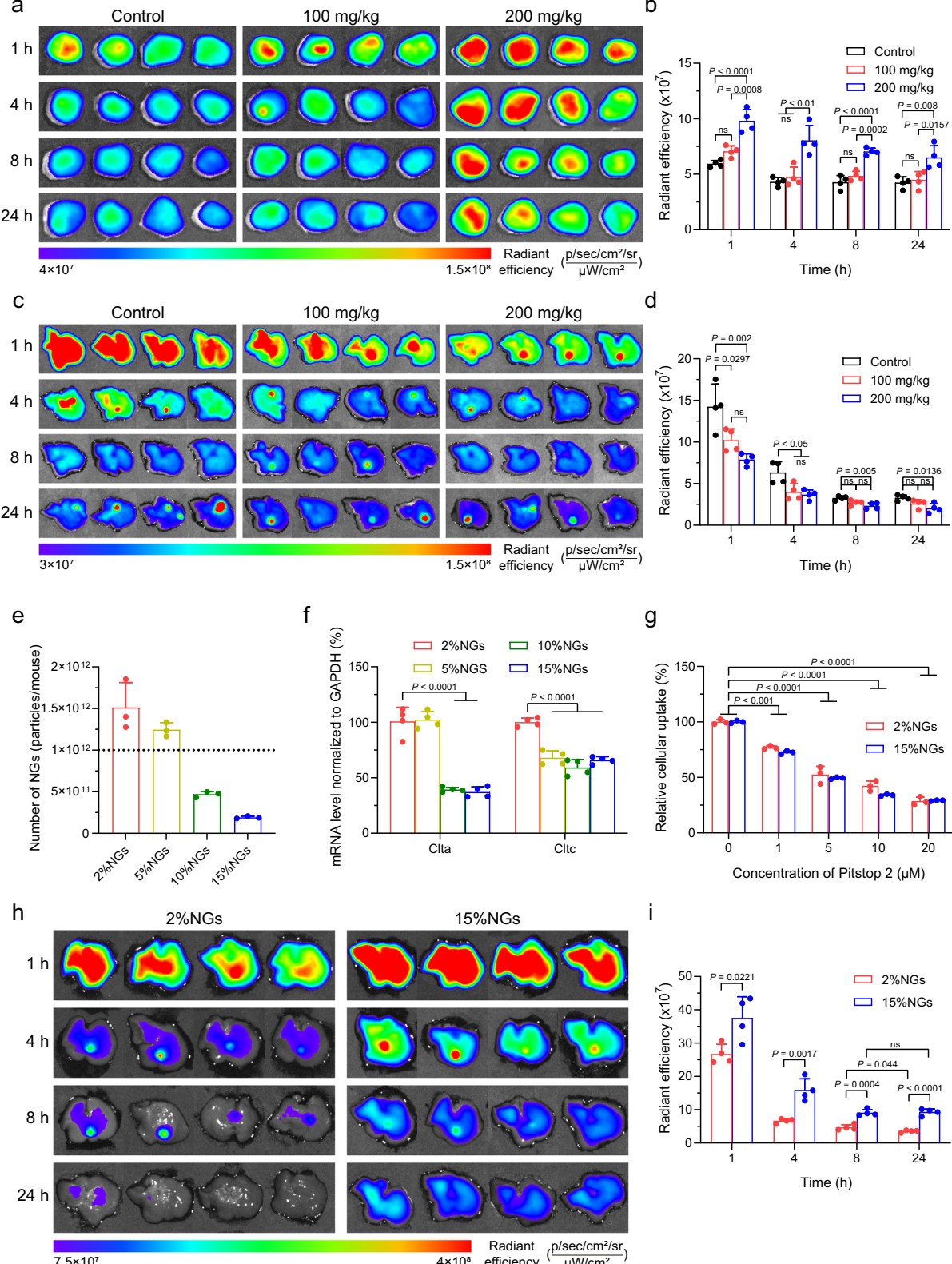

## Blockade by stiff nanogels boosting the antitumour efficacy of commercialized nanomedicine

Figure 6 revealed that RES-blockade by stiff nanogels could promote the antitumour efficacy of DOX-loaded nanogels, including both DOX@2%NGs and DOX@15%NGs. Next, we sought to explore whether stiffness-dependent RES-blockade could universally boost antitumour efficacy of commercialized nanomedicines, such as Doxil and

Abraxane. For this purpose, 4T1 subcutaneous tumor model was established and divided into seven groups including control (G1), Doxil (G2), 2%-blockade + Doxil (G3), 15%-blockade + Doxil (G4), Abraxane (G5), 2%-blockade + Abraxane (G6) and 15%-blockade + Abraxane (G7). The drug administration scheme is illustrated in Fig. 7a. Tumor volume and tumor weight results revealed that antitumour efficacy of both nanomedicine (G2 and G5) was obviously promoted by 15%-blockade

**Fig. 5 | Dosage- and stiffness-dependent RES-blockade.** Ex vivo fluorescent images of (**a**) tumor accumulation and (**c**) liver accumulation of ICG-loaded 2%NGs after RES-blockade by 15%NGs with different dosage. Semi-quantification of the quantity of ICG-loaded 2%NGs in (**b**) tumor and (**d**) liver. Data are presented as mean values ± SD (n = 4 biological independent replicates). **e** Number of nanogels with different stiffness for RES-blockade at dosage of 200 mg/kg. Dotted line indicates the number of $1 \times 10^{12}$ particles per mouse. Data are presented as mean values ± SD (n = 3 independent replicates). **f** Clathrin mRNA as normalized to GAPDH after incubation with nanogels with different stiffness. Data are presented as mean values ± SD (n = 4 biological independent replicates). **g** Relative cellular uptake of Rhodamine-labeled nanogels with different stiffness after inhibited by Pitstop 2 at different dosage in RAW 264.7 cells. Data are presented as mean values ± SD (n = 3 biological independent replicates). **h** Ex vivo fluorescent images of accumulation and retention of ICG-loaded nanogels with different stiffness in liver after injection. **i** Semi-quantification of the quantity of ICG-loaded nanogels with different stiffness in liver. Data are presented as mean values ± SD (n = 4 biological independent replicates). Statistical significance of (**b, d, f, g**) was calculated by one-way ANOVA. The statistical significance of (**i**) was calculated by unpaired two-sided Student's t test. Source data are provided as a Source Data file.

(G4 and G7) instead of 2%-blockade (G3 and G6). Doxil exhibited better antitumour efficacy than Abraxane with or without RES-blockade relying on better basic antitumour efficacy of Doxil (Fig. 7b–d). However, interestingly, after calculating volume- and weight-based tumor inhibition rate, we observed around 1.3-fold higher tumor inhibition rate for Doxil after 15%-blockade, whereas more than 2.5-fold higher tumor inhibition rate for Abraxane by 15%-blockade (Supplementary Fig. 29). Consistently, around 1.2-fold increase for DOX@2%NGs and 1.6-fold increase for DOX@15%NGs after 15%-blockade were observed (Supplementary Fig. 25). These results suggested that nanomedicines with lower antitumour efficacy would benefit more from RES-blockade with stiff nanogels. A similar conclusion was obtained that 200-nm AuNPs obtained 150-fold increase of tumor accumulation while 50-nm AuNPs obtained 20-fold increase, despite 50-nm AuNPs presented the highest tumor accumulation[25]. Also, after stained by H&E, sparser distribution of tumor cells could be observed in tumors after 15%-blockade than 2%-blockade (Supplementary Fig. 30). Besides, higher percentage apoptotic cells and lower percentage of proliferative cells were detected after blockade by stiff 15%NGs (Fig. 7f, g). A slight decrease of body weight in G4 might result from delayed clearance of Doxil after RES-blockade by 15%NGs (Fig. 7e). No obvious toxicity toward major organs was detected in H&E staining (Supplementary Fig. 31). Relatively higher ALT level was examined in G4 and G7, which were blocked by 15%NGs, similar to Supplementary Fig. 28a, in blood biochemical analysis and blood routine examine (Supplementary Fig. 32). However, all the indicators were in normal range and exhibited that commercialized nanomedicine combined with 15%-blockade caused negligible toxicity and side effects. Collectively, RES-blockade by stiff 15%NGs could boost antitumour efficacy of commercialized nanomedicines, demonstrating the universality and clinical translation potential of stiffness-dependent RES-blockade strategy in cancer therapy.

## Discussion

Herein, we designed and prepared a series of P(NIPMAM-ss-MAA) with distinctive stiffness by simply regulating the cross-linking degree. Utilizing these nanogels, we investigated how the mechanical properties affected drug delivery processes and found that stiff nanogels could inhibit clathrin-mediated endocytosis and remain in liver for a longer period than soft ones, while soft nanogels showed superior advantages in tumor accumulation, deep penetration, and antitumour efficacy. By taking full advantages of nanogel mechanical properties, we proposed a mechano-boosting strategy that combine the superiority of stiff nanogels in RES-blockade with the benefit of soft nanogels in drug delivery, namely stiff-blockade and soft delivery, for augmented antitumour efficacy (Fig. 6). Furthermore, we demonstrated translational potential of stiffness-dependent RES-blockade strategy in commercialized nanomedicines including both Doxil and Abraxane.

Nanomedicine mechanical properties have been demonstrated to play an essential role in tumor penetration and cellular internalization[36]. Stiffness-dependent drug delivery was verified in this work. Higher cellular uptake and cytotoxicity was observed in soft 2% NGs than stiff 15%NGs (Fig. 3b, c). Although in some nanoparticles, stiff ones showed higher cellular uptake efficiency than soft ones because of lower energy cost[67,68], many other parameters including structure, hydrophilicity, surface charge, internalization pathway, and range of Yong's modulus also affected endocytosis process. For example, soft HIV particles showed higher efficiency in entry into cells and soft HEMA nanogels showed higher cellular uptake efficiency by HepG2 cells[45,69]. It was remarkable that soft nanolipogels could enter cells via fusion endocytosis pathway because of similar components and structure to cell membrane while the stiff ones were mainly internalized via clathrin-mediated endocytosis pathway[63]. Besides, soft 2% NGs showed enhanced permeability (Fig. 3d, e) and deeper penetration distance away from blood vessels (Supplementary Fig. 10) than stiff 15%NGs, relying on better deformability to squeeze through dense extracellular matrix. As a result, soft 2%NGs showed higher tumor accumulation (Fig. 3f, g) and antitumour efficacy (Fig. 3h–j). As such, soft nanogels exhibited evident advantages in drug delivery, but they were still facing a severe problem that as extraneous nanoparticles they were rapidly cleared from blood circulation by RES[6,7]. Biodistribution showed that soft 2%NGs accumulated more in tumor, while stiff 15%NGs accumulated more in liver after 24 h post injection, leading to prolonged liver retention and liver clearance inhibition (Fig. 4a, b). Meanwhile, we found that after 1.5 h of incubation with 15% NGs, the cellular uptake of RAW 264.7 cells reach the largest amount (Supplementary Fig. 14), which was consistent with our previous work[12]. In combination of macrophage saturation and stationary tumor accumulation with time interval over 1.5 h, 1.5 h was set as time interval in this work. As illustrated in Fig. 4c–f, after blocked by 15%NGs, tumor accumulation was significantly enhanced and liver accumulation was inhibited. In striking contrast, no RES-blockade effect was observed for 2%NGs. These results supported that stiff 15%NGs could block RES more efficiently than soft 2%NGs, emphasizing that RES-blockade with nanoparticles is also stiffness-dependent.

Numerous factors exerted effects on RES-blockade including nanoparticle diameter, surface charge, and dosage. Nanoparticles with larger size could block RES more efficiently because of more opsonin adsorption for recognition and internalization by macrophages[3]. Similarly, positive charged liposomes achieved higher RES-blockade efficiency than negative charged liposomes relying on more effective electrostatic interactions with negative charged cell membrane[31]. Briefly, all these methods were established to increase the nanoparticle quantity internalized by macrophages, with the same purpose as blocking RES with high dosage. Herein, stiff 15%NGs showed dosage-dependent RES-blockade and the dosage of 200 mg/kg was identified as a threshold (Fig. 5a–d). However, as illustrated in Fig. 4c–f, soft 2% NGs hardly block RES at the same dosage. To quantify the relationship between dosage and liver clearance or tumor accumulation, 1 trillion particles per mouse had been demonstrated as a threshold for RES-blockade[35]. After calculation, the number of 2%NGs for RES-blockade was $(1.51 \pm 0.30) \times 10^{12}$ per mouse for RES-blockade which was higher than 1 trillion per mouse while 15%NGs was $1.91 \times 10^{11}$ per mouse which was much lower than 1 trillion per mouse (Fig. 5e). Stiff 15%NGs achieved higher RES-blockade efficiency compared to soft 2%NGs. To fully understand the relationship between RES-blockade efficiency and quantity of nanogels in macrophages, both cellular uptake and exocytosis were taken into consideration. Interestingly, we found that

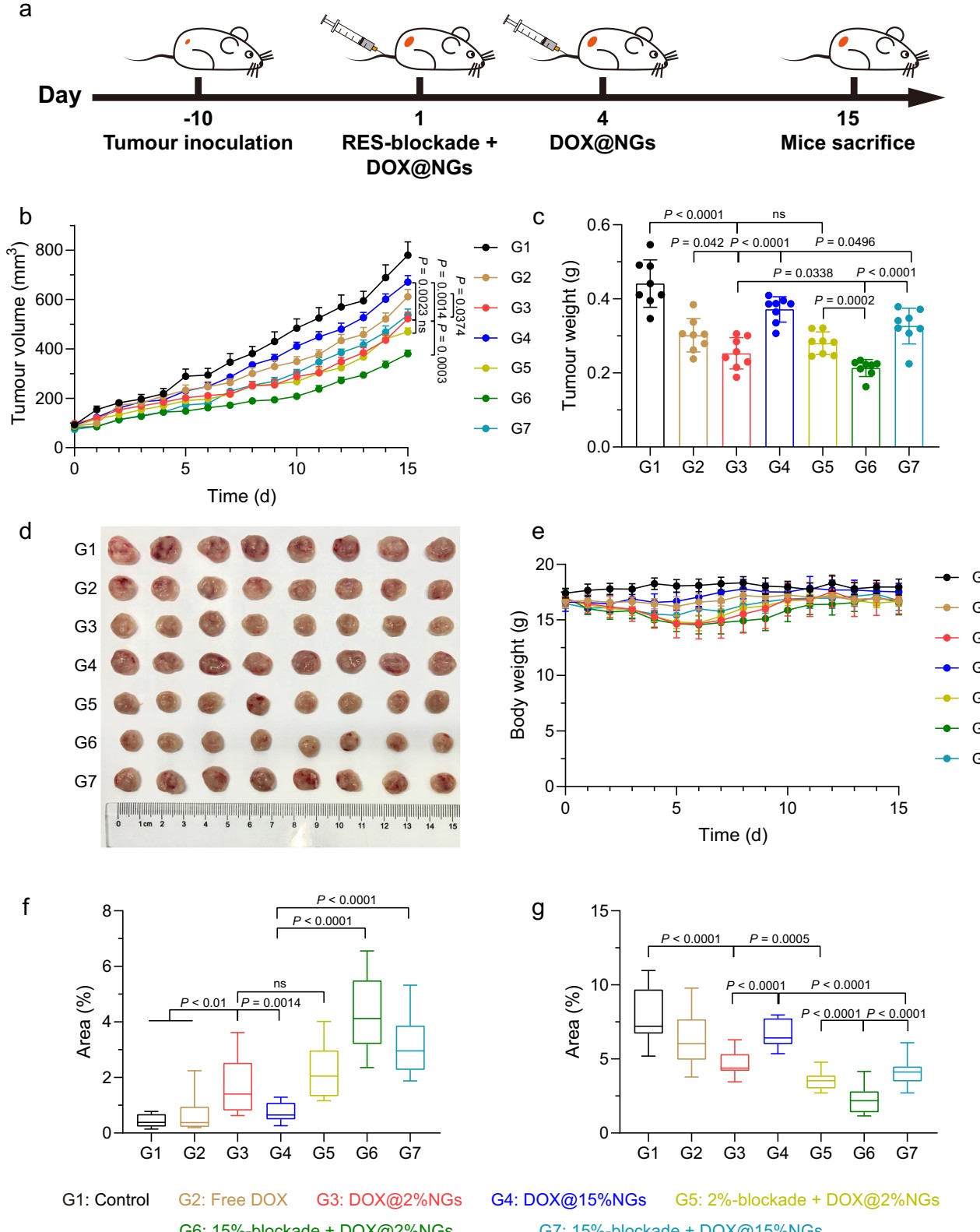

**Fig. 6 | Stiff-blockade combined with soft delivery contributing to enhanced antitumour efficacy. a** Illustration of drug administration strategy. **b** Tumor growth profiles of 4T1 tumor-bearing mice treated by different combination of RES-blockade and drug delivery strategy at RES-blockade dosage of 200 mg/kg and DOX dosage of 4 mg/kg. Data are presented as mean values ± SEM (*n* = 8 biological independent replicates). **c** Weight of tumors after treatment. Data are presented as mean values ± SD (*n* = 8 biological independent replicates). **d** Photograph of tumors extracted from the mice after treatment. **e** Body weight profiles of mice after treatment. Data are presented as mean values ± SD (*n* = 8 biological independent replicates). Percentage of the area of (**f**) apoptotic cells and (**g**) proliferate cells. Box plots indicate median (middle line), 25th, 75th percentile (box) and minimum and maximum (whiskers) (*n* = 15 independent replicates). Statistical significance was calculated by unpaired two-sided Student's *t* test. G1, Control; G2, Free DOX; G3, DOX@2%NGs; G4, DOX@15%NGs; G5, 2%-blockade + DOX@2%NGs; G6, 15%-blockade + DOX@2%NGs; G7, 15%-blockade + DOX@15%NGs. Source data are provided as a Source Data file.

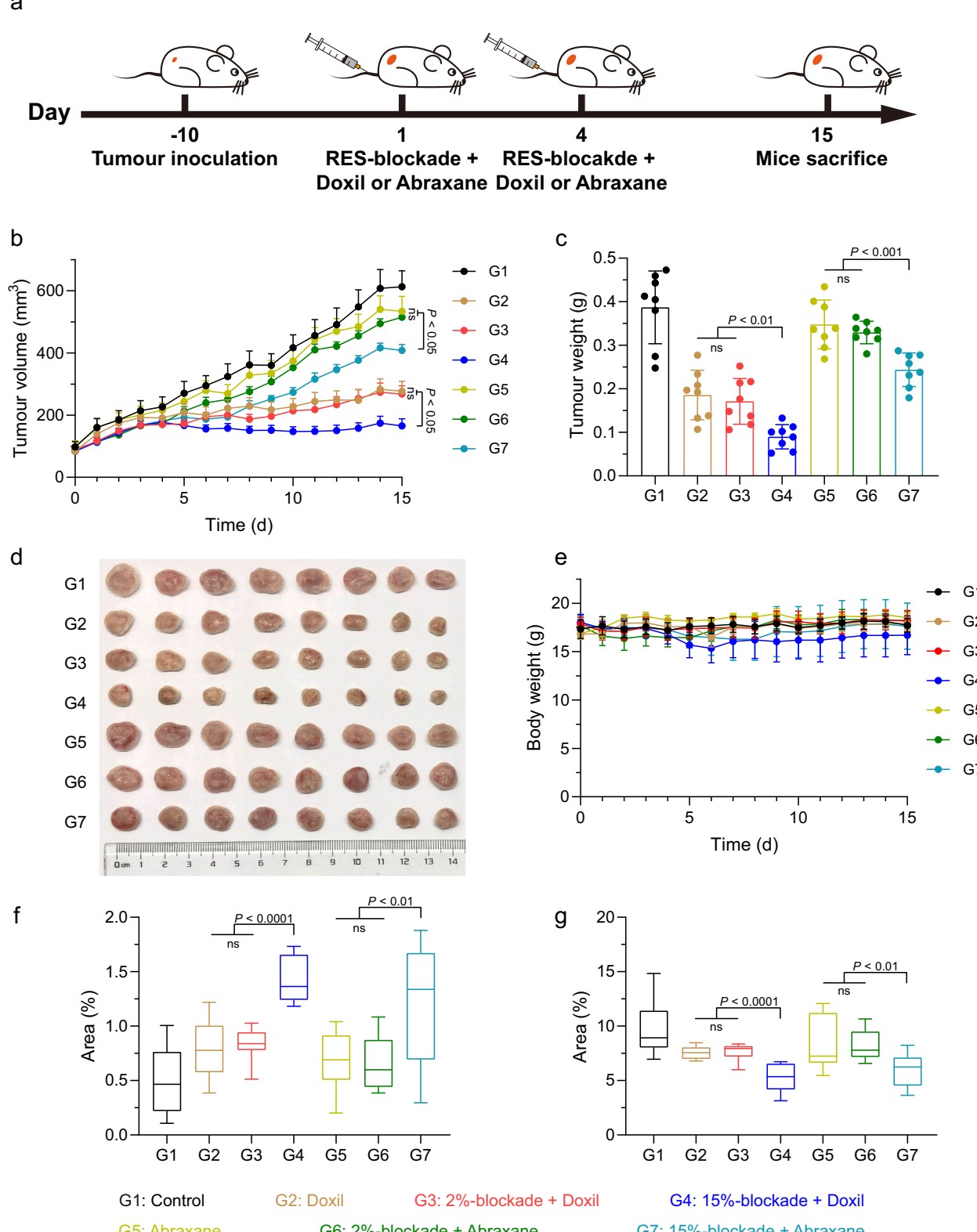

**Fig. 7 | Blockade by stiff nanogels boosting the antitumour efficacy of commercialized nanomedicine. a** Illustration of drug administration strategy. **b** Tumor growth profiles of 4T1 tumor-bearing mice treated by commercialized nanomedicine with or without RES-blockade by soft or stiff nanogels at blockade dosage of 200 mg/kg, DOX dosage of 3 mg/kg or paclitaxel dosage of 10 mg/kg. Data are presented as mean values ± SEM ($n = 8$ biological independent replicates). **c** Weight of tumors after treatment. Data are presented as mean values ± SD ($n = 8$ biological independent replicates). **d** Photograph of tumors extracted from the mice after treatment. **e** Body weight profiles of mice after treatment. Data are presented as mean values ± SD ($n = 8$ biological independent replicates). Percentage of the area of (**f**) apoptotic cells and (**g**) proliferate cells. Box plots indicate median (middle line), 25th, 75th percentile (box) and minimum and maximum (whiskers) ($n = 15$ independent replicates). Statistical significance was calculated by unpaired two-sided Student's *t* test. G1, Control; G2, Doxil; G3, 2%-blockade + Doxil; G4, 15%-blockade + Doxil; G5, Abraxane; G6, 2%-blockade + Abraxane; G7, 15%-blockade + Abraxane. Source data are provided as a Source Data file.

stiff 15%NGs blocking RES did not rely on large quantity of internalized nanogels by RAW 264.7 cells, because cellular uptake of 2%NGs was much higher than 15%NGs and similar efflux percentage of both 2%NGs and 15%NGs was observed (Supplementary Fig. 23), which meant fewer 15%NGs remained within RAW 264.7 cells but exhibited higher blockade efficiency (Fig. 4h). As such, the number or dosage was a key parameter but not the only one for RES-blockade, stiffness was also a vital parameter. In further investigation, we found that mRNA transcription level of clathrin and adapter protein AP-2 was downregulated after incubation with stiff 15%NGs rather than soft 2%NGs (Fig. 5f and Supplementary Fig. 24). After addition of clathrin-mediated endocytosis inhibitor Pitstop 2, the cellular uptake of both soft and stiff nanogels were significantly suppressed (Fig. 5g). Macrophages could recognize and uptake foreign nanoparticles via phagocytosis, macropinocytosis, and clathrin-mediated, caveolin-mediated, and clathrin/caveolin-independent endocytosis pathways[14]. Clathrin-mediated endocytosis has been demonstrated as the main internalization pathway of many nanoparticles. For example, hybrid nanoparticles with size up to 200 nm and nanogels with Young's modulus of 211 kPa were internalized via clathrin-mediated endocytosis[43,63]. This might be the reason why after incubation with stiff 15%NGs, the cellular uptake of subsequent nanoparticles was limited and why stiff 15%NGs showed lower internalization compared to soft 2%NGs (Supplementary Fig. 23c). In addition, longer retention time of 15%NGs in liver than 2%NGs (Fig. 5h, i) probably relied on the situation that stiff 15%NGs could hardly pass through liver sinusoidal fenestration or interact with hepatocytes and most of them were directly internalized by Kupffer cells[66]. In contrast, a larger quantity of soft 2%NGs could be cleared via hepatobiliary as a result of excellent deformability[6]. At the same time, dense structure of stiff 15%NGs resulted in slow biodegradation and sustaining RES-blockade efficacy while 2%NGs was quickly degraded by GSH after 24 h of incubation (Fig. 2g). To sum up, stiff 15%NGs achieved excellent RES-blockade capacity via inhibition of cell internalization and prolonged retention in the liver.

Clinical application was the final target of nanomedicines and nanotherapeutic strategy, especially for RES-blockade strategy which worked out mainly in vivo to prolong nanomedicine blood circulation and promote tumor accumulation. It has been demonstrated that blocking RES could suppress blood clearance of subsequent administered nanoparticles[34], and stiffness-dependent RES-blockade strategy enhanced antitumour efficacy of clinical nanomedicines Doxil and Abraxane (Fig. 7).

This work has four critical significances. First, the concept of mechano-boosting nanomedicine antitumour efficacy is established and nanogel mechanical properties are taken full advantage of to overcome pathophysiological barriers through the overall process of drug delivery, including blood circulation, tumor accumulation, tumor penetration, and cellular uptake. In detail, RES-blockade with stiff nanogels prolongs blood circulation of soft nanogels. Meanwhile, poor deformability of stiff nanogels results in limited permeability and low tumor accumulation to avoid unexpected tumor-blockade. Soft nanogels can penetrate deep site of tumor away from blood vessels and achieve efficient tumor accumulation as well as potent antitumour efficacy. Benefitting from the combined advantages of nanogels with distinctive stiffness at different drug delivery stages, the optimum antitumour effect is obtained. Second, nanogel stiffness as a parameter of RES-blockade is explored and we provide a plausible explanation to account for why stiff nanogels present higher RES-blockade efficiency than soft counterparts. On the one hand, stiff nanogels suppresses the ability of cellular uptake by downregulating mRNA transcription of clathrin and adapter protein AP-2. On the other hand, stiff nanogels present prolonged retention time in liver, ensuring stiff nanogels could sustain RES-blockade. Third, compared to depletion of macrophages, blocking with stiff nanogels is safer and more biocompatible. In addition, this process is reversible and temporary relying on

biodegradability of nanogels and time-dependent clearance from liver, thereby avoiding long-term unexpected side effects. Fourth, the promotion of commercialized nanomedicines antitumour efficacy after RES-blockade by stiff nanogels is also demonstrated. Considering that RES-blockade is clinical acceptable, the stiffness-dependent RES-blockade strategy has significant clinical translation potential to boost commercialized nanomedicines for cancer patients.

However, many issues call for further exploration. First, we have demonstrated stiff nanogels can suppress internalization of macrophages, but how stiff nanogels downregulate mRNA transcription of clathrin- or AP-related protein is not clear and the signaling pathways awaits further investigation. Second, whether nanogel stiffness also affects macrophages other biochemical properties, such as polarization, is unknown.

## Methods

### Materials
N-isopropylmethacrylamide (NIPMAM) and N,N'-bis(acryloyl)cystamine (BAC) were purchased from Sigma-Aldrich (St Louis, MO, USA). Methacrylic acid (MAA) was purchased from Sinopharm Chemical Reagent Co., Ltd. (Shanghai, China). Potassium persulfate (KPS) and sodium dodecyl sulfate (SDS) were purchased from Aladdin Reagent Inc. (Shanghai, China). Doxorubicin·HCl (DOX·HCl) was purchased from Meilun Biotech Co., Ltd. (Dalian, China). Indocyanine Green was purchased from J&K Scientific Ltd. (Beijing, China). Rhodamine B, 4-dimethylaminopyridine (DMAP), N,N'-Dicyclohexylcarbodiimide (DCC) and 2-hydroxyethyl methacrylate (HEMA) were purchased from Energy Chemical (Shanghai, China). Pitstop 2 was purchased from MedChemExpress (Shanghai, China). RNAprep FastPure, Goldenstar™ RT6 cDNA Synthesis Mix and Master qPCR Mix were purchased from Tsingke Biotechnology Co., Ltd. (Beijing, China).

### Cell culture and animals
The murine breast cancer cell line 4T1 was acquired from Shanghai Institutes for Biological Sciences. The NIH/3T3 cell line and the murine macrophage cell line RAW 264.7 were acquired from the National Collection of Authenticated Cell Cultures. The 4T1 cells and NIH/3T3 cells were cultured in RPMI 1640 medium and RAW 264.7 cells were cultured in Dulbecco's modified eagle medium (DMEM), supplemented with 10% fetal bovine serum and 1% antibiotics (penicillin: 100 U/mL, streptomycin: 100 μg/mL) at 37 °C under a 5% $CO_2$ atmosphere.

BALB/c mice (female) were purchased from Vital River Laboratory Animal Technology Co. Ltd. (Beijing, China). Mice were housed in an animal facility under constant environmental conditions (room temperature, $21 \pm 1$ °C; relative humidity, 40–70%, and a 12-h light–dark cycle). All mice had access to food and water ad libitum. All animal experiments were approved by the Institutional Animal Care and Use Committee at Tongji Medical College, Huazhong University of Science and Technology (Wuhan, China). The experiment protocols were approved by the Institutional Animal Ethical Committee of the Huazhong University of Science and Technology (Wuhan, China). The animal ethical clearance project number is 2019S924.

### Synthesis of P(NIPMAM-ss-MAA) nanogels with different stiffness
P(NIPMAM-ss-MAA) nanogels crosslinked by BAC were synthesized via emulsion polymerization. All the amount and molar ratio to NIPMAM of components are listed in Supplementary Table 3.

Monomer NIPMAM and MAA, surfactant SDS were dissolved in 80 mL of water in a three-necked round-bottom flask. Crosslinker BAC was pre-dissolved in ethyl alcohol and then added into the solution. The solution was vacuumized for 10 min to remove oxygen and ethyl alcohol, then inflated with argon. The operation was circulated for three rounds. The mixed solution was heated to 80 °C and the aqueous of KPS was injected into the solution to initiate the polymerization. The

reaction was maintained for 6 h under argon protection at 80 °C with stirring and then cooled to room temperature. The obtained nanogels were purified via ultrafiltration (molecular weight cutoff 10,000) to remove unpolymerized monomers and other small molecules and washed by water three times. Finally, the nanogels were concentrated to 20 mg/mL for storage at 4 °C, which was named as 2%NGs, 5%NGs, 10%NGs, 15%NGs.

### Preparation of DOX@Nanogels
In total, 5 mL of DOX·HCl aqueous solution with a concentration of 2 mg/mL was mixed with 5 mL of the obtained nanogel solutions and stirred for 48 h. The obtained DOX@Nanogels were purified via ultrafiltration (molecular weight cutoff 10,000) to remove free DOX. DOX@Nanogels were concentrated to 1 mg/mL of DOX and named as DOX@2%NGs, DOX@5%NGs, DOX@10%NGs, DOX@15%NGs. The concentration of DOX was determined on UV–vis spectroscopy at 483 nm.

### Preparation of ICG-loaded nanogels
In all, 5 mL of ICG aqueous solution with a concentration of 1 mg/mL was mixed with 5 mL of the obtained DOX@Nanogels solutions and stirred for 48 h. The obtained ICG-loaded nanogels were purified via ultrafiltration (molecular weight cutoff 10,000) to remove free ICG. ICG-loaded nanogels were concentrated to 1.5 mg/mL of ICG. The concentration of ICG was determined on UV–vis spectroscopy at 783 nm. In this work, ICG-loaded 2%NGs and ICG-loaded 15%NGs were used.

### Synthesis of Rhodamine-labeled nanogels
Overall, 5 g of Rhodamine B (10.45 mmol), 75 mg of DMAP (0.615 mmol), and 2.6 g of DCC (12.6 mmol) were dissolved in 52.5 mL of anhydrous $CH_2Cl_2$ and the solution was vacuumed to remove oxygen. After stirring for 30 min under argon protection, 1.55 mL HEMA (12.5 mmol) was added. The mixed solution was stirred at 20 °C for 25 h under argon protection. The reaction product was purified by column chromatography on silica gel with 90/10 DCM/MeOH eluent and concentrated on rotary evaporator to obtain dry purple powder (RhB-HEMA).

Similar to the synthesis of P(NIPMAM-ss-MAA), 255 μg of RhB-HEMA with molar ratio of 5% to monomer NIPMAM was added into water together with other monomers, followed by the same operation. Also, Rhodamine-labeled nanogels were concentrated to 20 mg/mL for storage at 4 °C.

### Characterization of P(NIPMAM-ss-MAA) nanogels
The hydrodynamic diameter distribution and zeta potentials of nanogels were measured by dynamic light scattering (DLS, Malvern, Zetasizer Nano-ZS, UK) at 37 °C, with diameters measured in PBS buffer and zeta potentials in water. The data was collected with Zetasizer Software (Ver. 7.13).

The morphologies of nanogels were characterized by transmission electron microscopy (TEM, JEOL, JEM-1230, Japan). In total, 10 μL of nanogel solution (0.01 mg/mL) was dropped onto a 400-mesh carbon-coated copper grid. After drying in air for 24 h, the nanogels were stained with phosphomolybdic acid for 2 min and washed by water once. Then the nanogels were characterized on a TEM after drying.

Young's modulus was determined by atomic force microscopy (AFM, Bruker, Multimode 8, Germany) in liquid phase. The mica sheet was soaked in polyethyleneimine (PEI) solutions for modification with positive charge. In total, 100 μL of nanogel solutions were dropped onto the sheet and maintained for 10 min to ensure adsorption onto the sheet. Then the force curve was measured in solution with tapping mode and Young's modulus was further calculated with NanoScope Analysis (Ver. 1.5) software.

Temperature-responsiveness was measured by DLS under trend mode from 25 to 55 °C and equilibration time of each temperature point was 1 min. The nanogel solutions was diluted with water at different pH from 3 to 9 and pH-responsiveness was measured by DLS. 2% NGs was incubated with GSH solutions (10 mmol/L) for 24 h and redox-responsiveness was characterized by TEM relying on the morphology variation.

Number concentration was determined by nanoparticle tracking analysis (NTA, Malvern, NanoSight NS300, UK). The nanogels with different stiffness were diluted to appropriate concentration with PBS buffer and loaded into the sample chamber at ambient temperature. Three 60-second videos were acquired for each sample. The videos were analyzed with NanoSight NTA (Ver. 3.3) software. The number of nanogels per milligram and the number of nanogels injected per mouse was calculated accordingly.

### Characterization of DOX@Nanogels
Hydrodynamic diameter distribution and temperature-responsiveness was measured by DLS. Morphology was characterized by TEM and AFM. Young's modulus was measured by AFM. Stability in PBS and FBS was evaluated for 4 days, according to the hydrodynamic diameter on each day by DLS.

### MTT assay in 4T1 cells
The biocompatibility of blank nanogels was evaluated by MTT assay. 4T1 cells were seeded in 96-well plates at $5 \times 10^3$ cells/well and incubated for 12 h. Medium was removed and serum-free medium containing different concentration of blank nanogels with different stiffness was added to the wells. After incubation for 24 h, the medium containing nanogels was removed and new serum-free medium containing 20 μL of 5 mg/mL MTT was added for additional 4 h incubation at 37 °C. The medium containing MTT was removed and 150 μL of DMSO was added to dissolve the formazan crystals. The absorption value was measured by microplate reader (Molecular Devices, Flex Station 3, USA) at 570 nm and cell viability was calculated.

DOX@2%NGs and DOX@15%NGs were used to evaluate the stiffness-dependent cytotoxicity by MTT assay. Similar to the above operation, after cells were seeded for 12 h, serum-free medium containing DOX@2%NGs or DOX@15NGs with different concentrations of DOX was added for another 24 h incubation. After 4 h of incubation with MTT, the formazan crystals were dissolved by DMSO. The light absorption value was measured by microplate reader at 570 nm and cell viability was calculated.

### In vitro cellular uptake
Rhodamine-labeled nanogels with different stiffness were used to evaluate the cellular uptake efficiency. 4T1 cells were seeded in 6-well plates at $5 \times 10^5$ cells/well and incubated for 12 h. Medium was removed and serum-free medium containing 50 μg/mL of Rhodamine-labeled nanogels was added to the well for further 4 h of incubation. After being washed by PBS three times, the cells were collected for flow cytometry analysis (Berkman Coulter, CytoFLEX, USA). The data were analyzed with CytExpert (Ver. 2.4), and relative cellular uptake efficiency was calculated.

Rhodamine-labeled 15%NGs was used to investigate the cellular uptake efficiency by RAW 264.7 cells at different time point. RAW 264.7 cells were seeded in six-well plates at $5 \times 10^5$ cells/well and incubated for 12 h. Medium was removed and serum-free medium containing 50 μg/mL of Rhodamine-labeled 15%NGs was added to the well. After incubation for 0.5, 1, 2, 4, and 6 h, the cells were washed by PBS three times and collected for flow cytometry analysis.

Rhodamine-labeled nanogels with different stiffness were used to evaluate the cellular uptake efficiency. RAW 264.7 cells were seeded in six-well plates at $5 \times 10^5$ cells/well and incubated for 12 h. Medium was removed and serum-free medium containing 50 μg/mL of Rhodamine-

labeled nanogels was added to the well for further 1.5 h of incubation. After being washed by PBS three times, the cells were collected for flow cytometry analysis and relative cellular uptake efficiency was calculated.

Rhodamine-labeled 2%NGs and Rhodamine-labeled 15%NGs were used to evaluate the exocytosis efficiency. RAW 264.7 cells were seeded in six-well plates at $5 \times 10^5$ cells/well and incubated for 12 h. Medium was removed and serum-free medium containing 50 μg/mL of Rhodamine-labeled nanogels was added to the well for further 1.5 h of incubation. The medium was removed and the cells were washed by PBS three times. New medium was added and incubated for 1.5 h or 6 h. After being washed by PBS three times, the cells were collected for flow cytometry analysis and the remained relative fluorescent intensity in cells was calculated.

2%NGs and 15%NGs were used to evaluate stiffness-dependent RES-blockade efficiency. RAW 264.7 cells were seeded in 6-well plates at $5 \times 10^5$ cells/well and incubated for 12 h. Medium was removed and new medium containing 400 μg/mL of 2%NGs or 15%NGs were added to the cell. After incubation for 1.5 h, medium containing 50 μg/mL of Rhodamine-labeled 2%NGs were added into the well for another 1.5 h of incubation. The cells were washed by PBS three times and collected for flow cytometry analysis.

In all, 15%NGs was used to evaluate dosage-dependent RES-blockade efficiency. RAW 264.7 cells were seeded in 6-well plates at $5 \times 10^5$ cells/well and incubated for 12 h. Medium was removed and serum-free medium containing 100 μg/mL, 200 μg/mL, 300 μg/mL, 400 μg/mL of 15%NGs were added to the cell. After incubation for 1.5 h, serum-free medium containing 50 μg/mL of Rhodamine-labeled 2% NGs were added into the well for another 1.5 h of incubation. The cells were washed by PBS three times and collected for flow cytometry analysis.

### mRNA transcription level of RAW 264.7 cells

RAW 264.7 cells were pretreated with nanogels of different stiffness at a concentration of 100 μg/mL for 24 h. Then, RNA of RAW 264.7 cells was isolated with RNAprep FastPure. RNA was eluted in $H_2O$ and reverse-transcribed to cDNA following the protocol of Goldenstar™ RT6 cDNA Synthesis Mix. cDNA samples were run on StepOnePlusTM real-time PCR system using Master qPCR Mix. The forward primer and reverse primer of each gene for qPCR are listed in Supplementary Table 4.

Pitstop 2 was used as clathrin-mediated endocytosis inhibitor to investigate the cellular uptake efficiency of RAW 264.7 cells after suppressing clathrin-mediated endocytosis. RAW 264.7 cells were seeded in six-well plates at $5 \times 10^5$ cells/well and incubated for 12 h. Medium was removed and serum-free medium containing 1 μM, 5 μM, 10 μM, 20 μM of Pitstop 2 was added to the well for 0.5 h of incubation. Serum-free medium containing 50 μg/mL of Rhodamine-labeled 2% NGs or Rhodamine-labeled 15%NGs was added to the well for further 1.5 h of incubation. After being washed by PBS three times, the cells were collected for flow cytometry analysis and relative cellular uptake efficiency was calculated.

### In vitro stiffness-dependent penetration

In all, 1 mL of Matrigel was added to the bottom of 1 mL EP tube. In total, 100 μg/mL of Rhodamine-labeled 2%NGs or Rhodamine-labeled 15%NGs was added to the top of Matrigel. After 6 hours of incubation, the nanogels were removed and the depth of penetration in Matrigel could be observed.

NIH/3T3 cells and 4T1 cells were mixed in the ratio of 1:2 in medium with 0.24% of methylcellulose. $1.5 \times 10^4$ cells in 25 μL medium was seeded on the internal surface of a culture dish lid. Then the lid was flipped to make the medium drop suspended and covered on the culture dish. 3D tumor spheroids formed after being cultured for 72 h. The 3D tumor spheroids were transferred into new medium with 100 μg/mL of Rhodamine-labeled 2%NGs or Rhodamine-labeled 15% NGs for another 6 h of culturing. The 3D tumor spheroids were washed by PBS three times and the distribution of fluorescence was characterized by confocal laser scanning microscope (Olympus, FV3000, Japan). The images were analyzed with FluoView31S (Ver. 2.3) software.

### In vivo and ex vivo stiffness-dependent tumor accumulation

4T1 tumor model was established via subcutaneous injection of $1 \times 10^6$ 4T1 cells in 100 μL of PBS. When the tumor volume reached around 200 $mm^3$, the mice were randomly divided into ICG-loaded 2%NGs group and ICG-loaded 15%NGs group. After drug administration at ICG dosage of 4 mg/kg via i.v. injection, the mice were anesthetized and imaged at 1, 2, 4, 8, 12, and 24 h by in vivo imaging system (PerkinElmer, IVIS Lumina XR, USA). At 24 h, the mice were sacrificed and the tumors and major organs were harvested for imaging. The images were analyzed with Living Image (Ver. 4.0) software.

For ex vivo imaging, the mice were sacrificed, tumors and organs were harvested at 1, 4, 8, and 24 h after drug administration.

To further investigate the tumor accumulation relying on stiffness of nanogels, both Rhodamine-labeled 2%NGs and 15%NGs were used for ex vivo imaging. 4T1 tumor model was established as described above. When the tumor volume reached around 200 $mm^3$, the mice were randomly divided into two groups including Rhodamine-labeled 2%NGs, Rhodamine-labeled 15%NGs. The two groups were i.v. injected different nanogels with same fluorescent intensity because of the inaccurate quantification of Rhodamine B affected by chemical coupling. After 4 h of injection, the mice were sacrificed and the tumors were harvested for imaging by IVIS.

### In vivo time-dependent RES-blockade efficiency

4T1 tumor model was established as described above. When the tumor volume reached around 200 $mm^3$, the mice were randomly divided into four groups, including control group, 0.5 h group, 1.5 h group and 3 h group. Control group was i.v. injected saline, while 2%-blockade groups was i.v. injected 2%NGs and 15%-blockade groups was i.v.-injected 15%NGs at a dosage of 200 mg/kg. At 0.5 h, 1.5 h or 3 h after injection, ICG-loaded 2%NGs was i.v. injected at ICG dosage of 4 mg/kg. After administration of the mice were anesthetized and imaged at 1, 2, 4, 8, 12, and 24 h by IVIS, including tumor site and liver site.

### In vivo and ex vivo stiffness-dependent RES-blockade efficiency

4T1 tumor model was established as described above. When the tumor volume reached around 200 $mm^3$, the mice were randomly divided into three groups including control group, 2%-blockade group and 15%-blockade group. Control group was i.v.-injected saline, while 2%-blockade group was i.v. injected 2%NGs and 15%-blockade group was i.v. injected 15%NGs at a dosage of 200 mg/kg. At 1.5 h after injection, ICG-loaded 2%NGs was i.v. injected at ICG dosage of 4 mg/kg. After administration of ICG the mice were anesthetized and imaged at 1, 2, 4, 8, 12, and 24 h by IVIS, including tumor site and liver site.

For ex vivo imaging, the mice were sacrificed, tumors and livers were harvested at 1, 4, 8, and 24 h after drug administration.

### In vivo and ex vivo dosage-dependent RES-blockade efficiency

4T1 tumor model was established as described above. When the tumor volume reached around 200 $mm^3$, the mice were randomly divided into four groups, including control group, 100 mg/kg group, 200 mg/kg group and 300 mg/kg group. Control group was i.v. injected saline, while other groups were i.v. injected 15%NGs at a dosage of 100 mg/kg, 200 mg/kg, 300 mg/kg. At 1.5 h after injection, ICG-loaded 2%NGs was i.v. injected at ICG dosage of 4 mg/kg. After drug administration of ICG the mice were anesthetized and imaged at 1, 2, 4, and 10 h by IVIS, including tumor site and liver site.

For ex vivo imaging, the mice were sacrificed, tumors and livers were harvested at 1, 4, 8, and 24 h after drug administration.

## Pharmacokinetics after RES-blockade

The mice were divided into 3 groups including control, 2%-blockade and 15%-blockade. Control group was i.v. injected saline, while 2%-blockade group was i.v. injected 2%NGs and 15%-blockade group was i.v. injected 15%NGs at a dosage of 200 mg/kg. At 1.5 h after injection, ICG-loaded 2%NGs was i.v. injected at ICG dosage of 6 mg/kg. In all, 80 μL of blood was collected via tail vein and added to tubes with 10 μL of EDTA-K. The samples were centrifuged at 278×$g$ for 10 min. Overall, 10 μL of plasma was collected and diluted with 50 μL of DMSO, then detected by microplate reader at excitation wavelength of 783 nm. Concentration of ICG in blood and % I.D. were calculated. Pharmacokinetic parameters were calculated with Data Analysis System (Ver. 2.0) software.

## In vivo stiffness-dependent antitumour efficacy

4T1 tumor model was established as described above. When the tumor volume reached around 100 mm$^3$, the mice were randomly divided into five groups including control, free DOX, DOX@2%NGs, DOX@10%NGs, DOX@15%NGs. Control group was i.v. injected saline at day 1 and day 4 while other groups were respectively i.v. injected free DOX, DOX@2%NGs, DOX@10%NGs, DOX@15%NGs at DOX dosage of 4 mg/kg. The tumor length and width were measured by a digital vernier caliper and body weight was measured by an electronic scale every day for 16 days. The tumor volume was calculated according to the formula: tumor volume = length × width$^2$/2. At the end of treatment, the mice were sacrificed and tumors were harvested for weighing and photos. Then the tumors were fixed with 4% paraformaldehyde for H&E staining, TUNEL staining and Ki67 (1:200 for dilution) staining to evaluate necrosis, apoptosis, and proliferation of tumor cells. Percentage of apoptosis and proliferation area was semi-quantified by the software ImageJ (Ver. 2.0). Besides, the tumor was staining with fluorescence-labeled CD31 antibody (1:50 for dilution) to evaluate penetration efficiency of nanogels with different stiffness. Meanwhile, major organs including heart, liver, spleen, lung and kidneys were harvested and fixed for H&E staining to evaluate the histological toxicity of each group. Besides, the blood of mice was collected for blood biochemical analysis and blood routine examine to evaluate toxicity.

## In vivo antitumour efficacy of different RES-blockade strategies

4T1 tumor model was established as described above. When the tumor volume reached around 100 mm$^3$, the mice were randomly divided into seven groups including control, free DOX, DOX@2%NGs, DOX@15%NGs, 2%-blockade + DOX@2%NGs, 15%-blockade + DOX@2%NGs, 15%-blockade + DOX@15%NGs. Control group was i.v. injected saline at day 1 and day 4. Free DOX, DOX@2%NGs, DOX@15% NGs groups were respectively i.v. injected free DOX, DOX@2%NGs, DOX@15%NGs at DOX dosage of 4 mg/kg at day 1 and day 4. 2%-blockade + DOX@2%NGs group was i.v. injected 2%NGs at a dosage of 200 mg/kg and DOX@2%NGs at DOX dosage of 4 mg/kg 1.5 h later at day 1, and only DOX@2%NGs at day 4. 15%-blockade + DOX@2%NGs group was i.v. injected 15%NGs at a dosage of 200 mg/kg and DOX@2% NGs at DOX dosage of 4 mg/kg 1.5 h later at day 1, and only DOX@2% NGs at day 4. 15%-blockade + DOX@15%NGs group was i.v. injected 15% NGs at a dosage of 200 mg/kg and DOX@15%NGs at DOX dosage of 4 mg/kg 1.5 h later at day 1, and only DOX@15%NGs at day 4. The tumor volume was calculated and body weight were measured every day for 16 days. At the end of treatment, the mice were sacrificed and tumors were harvested for weighing and photos. Then the tumors were fixed with 4% paraformaldehyde for H&E staining, TUNEL staining and Ki67 staining to evaluate necrosis, apoptosis, and proliferation of tumor cells. The percentage of apoptosis and proliferation area was semi-quantified by the software ImageJ. Meanwhile, major organs including heart, liver, spleen, lung and kidneys were harvested and fixed for H&E staining to evaluate the histological toxicity of each group. Besides, the blood of mice was collected for blood biochemical analysis and blood routine examine to evaluate toxicity.

## In vivo antitumour efficacy of commercial nanomedicine after RES-blockade

4T1 tumor model was established as described above. When the tumor volume reached around 100 mm$^3$, the mice were randomly divided into seven groups including control, Doxil, 2%-blockade + Doxil, 15%-blockade + Doxil, Abraxane, 2%-blockade + Abraxane, 15%-blockade + Abraxane. Control group was i.v. injected saline at day 1 and day 4. Doxil and Abraxane groups were, respectively, i.v. injected Doxil at DOX dosage of 3 mg/kg or Abraxane at PTX dosage of 10 mg/kg at day 1 and day 4. 2%-blockade + Doxil and 2%-blockade + Abraxane groups was i.v. injected 2%NGs at a dosage of 200 mg/kg and Doxil at DOX dosage of 3 mg/kg or Abraxane at PTX dosage of 10 mg/kg 1.5 h later at day 1 and day 4. 15%-blockade + Doxil and 15%-blockade + Abraxane groups was i.v. injected 15%NGs at a dosage of 200 mg/kg and Doxil at DOX dosage of 3 mg/kg or Abraxane at PTX dosage of 10 mg/kg 1.5 h later at day 1 and day 4. The tumor volume was calculated and body weight were measured every day for 16 days. At the end of treatment, the mice were sacrificed and tumors were harvested for weighing and photos. Then the tumors were fixed with 4% paraformaldehyde for H&E staining, TUNEL staining and Ki67 staining to evaluate necrosis, apoptosis, and proliferation of tumor cells. The percentage of apoptosis and proliferation area was semi-quantified by the software ImageJ. Meanwhile, major organs including the heart, liver, spleen, lung, and kidneys were harvested and fixed for H&E staining to evaluate the histological toxicity of each group. Besides, the blood of mice was collected for blood biochemical analysis and blood routine examine to evaluate toxicity.

## Statistical analysis

Statistical analysis was performed using GraphPad Prism 8.0 software. Data were presented as mean values ± SD or mean values ± SEM. Statistical significance was calculated by unpaired two-sided Student's $t$ test between two groups and one-way ANOVA for comparison of multiple groups. $P$ values of <0.05 were considered statistically significant.

## Reporting summary

Further information on research design is available in the Nature Portfolio Reporting Summary linked to this article.

# Data availability

The authors declare that data supporting the findings of this study are available within the Article, Supplementary Information and Source Data File. Source data are provided with this paper.

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

## Acknowledgements

We thank the Research Core Facilities for Life Science (HUST), the Optical Bioimaging Core Facility of WNLO-HUST, and the Analytical and Testing Center of HUST for the facility support. This work was financially supported by grants from the National Research and Development Program of China (2018YFA0208900 (Z.F.L.), 2020YFA0211200 (Z.F.L.), and 2020YFA0710700 (X.L.Y.)), the National Science Foundation of China (82172757 (Z.F.L.), 31972927 (Z.F.L.)), the Scientific Research Foundation of Huazhong University of Science and Technology (3004170130) (Z.F.L.), the Program for HUST Academic Frontier Youth Team (2018QYTD01) (Z.F.L.), and the HCP Program for HUST (X.L.Y.).

## Author contributions

Z.F.L. and X.L.Y. designed the project. Z.L., Y.B.Z., H.W.Z., C.W., C.X., Q.W., H.M.W., S.Y.L., J.T.C., and C.X. performed the experiments. Z.L. and Z.F.L. analyzed and interpreted the data. Z.L. and Z.F.L. wrote and revised the manuscript.

## Competing interests

The authors (Z.F.L., X.L.Y., and Z.L.) have applied for patents related to this study. The remaining authors declare no competing interests.
