## [Peer Review File · Nature Communications]

Mechano-boosting nanomedicine antitumour efficacy by blocking the reticuloendothelial system with stiff nanogelsReviewers' Comments:

Reviewer #1:

Remarks to the Author:

This manuscript reports a study investigating the stiffness of nanogels on the RES clearance. The RES clearance has been a challenge for tumor targeting nanoparticles. The authors prepared nanogels with different stiffness and reported that soft nanogels are excellent for tumor targeting while stiff nanogel may be used for RES blocking and combined chemotherapy with commercial drugs. The physicochemical properties influence substantially the in vivo fate of nanoparticles. The effect of the stiffness of nanoparticles on macrophage phagocytosis has been reported. While the impact of stiffness of nanogel as nanomedicine has not been understood, the finding of the study may not be particularly new. Specific comments are provided below.

1. The DLS result shows similar sizes for the four nanogels; however, from the TEM images and Figure S1C, it seems they have very different size distributions from 200 to 350 nm. The size is an important factor in this study as it may affect the blocking effects, drug loading, cellular uptake, and therapeutic effects. The authors need to provide more data concerning the sizes of nanogels and make sure they are similar in all characterization experiments.
2. From the IVIS images, the area with fluorescence is significantly larger than actually the tumor site and nearly all areas without mice fur cover showed fluorescent signals (Figure 3F; Figure 4C; Figure 5A), which means the detection level may be different for the time points. The authors need to reorganize the graphs and provide ex vivo fluorescence images.
3. The RES has two major effects on nanoparticles: blood clearance and tumor accumulation. The authors focused mainly on the tumor accumulation part, while the circulation remains largely uninvestigated. The inclusion of the pharmacokinetic study would be beneficial for presenting the RES blockade effects.
4. Figure 1. Captions in the figure are required for a better illustration of what the study is mainly about.
5. For cell viability tests, incubation with a longer time than 24h is suggested for better showing the safety of the nanogel.
6. For quantitative analysis of cellular uptake, flow cytometry graphs and data should be provided.
7. For the biodistribution study, it is shown that, during 1~4h, the nanogels also distribute significantly in other organs, it would be interesting to investigate where the nanogels accumulate instead of the liver and tumor.
8. In the Introduction section, when mentioning the modification method, the authors need to cite appropriate evidence for the statement: "potential immune suppress induced by CD47-derived peptide will cause unexpected tumor growth or metastasis".

Reviewer #2:

Remarks to the Author:

This manuscript is addressing a strategy to change biodistribution of intravenously injected nanoparticles by inhibiting nanoparticle clearance through macrophage blockade. To inhibit the clearance, saturation of phagocytosis was induced by prompting macrophage endocytosis of stiff nanogels. Using this strategy, authors achieved redistribution of soft nanogels loaded with anticancer drug, which then led to improvement of anticancer effect in mouse model. Applicability of approach for FDA-approved nanoparticles was also demonstrated. While the overall concept of RES blockade is interesting and appealing to researchers working in the nanomedicine space, there are a several concerns with this study:

Major comments:

- 1) Drug loading can change colloidal stability of nanoparticles, as well as other their properties. Authors investigated this issue only for 2% NGs. The same studies must be performed for 15% NGs, which is widely used in the manuscript.
- 2) Treatments with nanogels lead to a drop of mice weight by several grams, with no returning for

normal values. It indicates toxicity, which should be explained and described. Nevertheless, authors mentioned that only free dox have cardiotoxicity, refereeing on Fig 3L, which is not obvious from this graph. I should add that the same weight loss can be observed in Figure 6E.

3) In blood biochemical parameters, several indicators significantly decreased (see, for example, WBC: Figures S19, S23). Even if they are in normal range, these accidents should be mentioned in the text and discussed, as possible signs of toxicity.

4) Time interval for RES blockade in vivo was determined from the in vitro experiments. Since nanoparticle behavior in mice bloodstream is more complex than interaction of nanoparticles with macrophages in vitro, this study raises serious concerns: What is the blood circulation time of 2% and 15% NGs in vivo? Why was the cellular uptake determined only for 15% NGs? Can the time interval be different for nanogels with different stiffness? Authors should rationalize their injection regime by experiments on mice.

5) Analysis of variance (ANOVA) is preferable for statistical analyses between three or more groups to avoid type 1 error.

6) Authors claimed suppression of clathrin-mediated endocytosis as a major mechanism of blockade. Nevertheless, this mechanism was defined from in vitro studies and can be non-sole in vivo. Authors should increase discussion about other possible mechanism which can cause observed blockade.

Minor comments:

1) Line 45-46: Add than in average, 0.7% was delivered.

2) Lines 49-50: Authors suppose that RES = MPS, which is not correct since these systems include different types of cells. They should clarify which system they blockade – if it only macrophages, it rather MPS blockade then RES blockade.

3) Introduction: a lot of recent achievements in the macrophage blockade field were ignored, like nanoprimers technology (Saunders et al, Nano letters, 2020); cyto blockade with RBCs (Nikitin et al, Nat. Biomed. Eng., 2020) or exosomes blockade (Belhadj, J. extracellular vesicles, 2020).

4) Lines 253-255: specify is it mean tumor volumes? Add SD

5) Fig 4ac,d and other bioimaging – add quantitative scale diapason

6) Fig S13 – correct typo “particals”

Point-by-point response to the reviewers' comments

We really appreciate all reviewers for their insightful and constructive comments. Below, the comments are shown in blue, and our responses are in standard typeface. Sections highlighted in yellow represent wording that has been altered/inserted into the revised manuscript. The corresponding sections in the revised manuscript itself are also highlighted in yellow.

Reviewer #1: This manuscript reports a study investigating the stiffness of nanogels on the RES clearance. The RES clearance has been a challenge for tumor targeting nanoparticles. The authors prepared nanogels with different stiffness and reported that soft nanogels are excellent for tumor targeting while stiff nanogel may be used for RES blocking and combined chemotherapy with commercial drugs. The physicochemical properties influence substantially the in vivo fate of nanoparticles. The effect of the stiffness of nanoparticles on macrophage phagocytosis has been reported. While the impact of stiffness of nanogel as nanomedicine has not been understood, the finding of the study may not be particularly new. Specific comments are provided below.

Our response: We thank this reviewer for the valuable comments. We will try our best to address the comments.

1. The DLS result shows similar sizes for the four nanogels; however, from the TEM images and Figure S1C, it seems they have very different size distributions from 200 to 350 nm. The size is an important factor in this study as it may affect the blocking effects, drug loading, cellular uptake, and therapeutic effects. The authors need to provide more data concerning the sizes of nanogels and make sure they are similar in all characterization experiments.

Our response: We really appreciate the reviewer for pointing out the problem. In this work, we designed and prepared nanogels with distinctive stiffness and ensured similar size distribution of nanogels in PBS at 37 °C to simulate physiological conditions.

For TEM images, the samples were dried during preparation and nanogels would shrink because the network structure would collapse after removing H₂O among polymer

networks. In addition, 2%NGs showed stronger shrinkage relying on lower crosslinking density while 15%NGs presented weaker shrinkage, so wider size distribution could be observed in TEM images. Figure S1C was pH-responsiveness of nanogels in H₂O and the strongest swelling could be observed in 2%NGs, similarly.

In cell experiments and *in vivo* experiments, the influence coming from diameter of nanogels could be avoided, including cellular uptake, RES-blockade, biodistribution and pharmacodynamics study because of similar size distribution of nanogels under physiological conditions. At the same time, the drug-loading capacity of nanogels with different stiffness was well controlled at around 5%.

2. From the IVIS images, the area with fluorescence is significantly larger than actually the tumor site and nearly all areas without mice fur cover showed fluorescent signals (Figure 3F; Figure 4C; Figure 5A), which means the detection level may be different for the time points. The authors need to reorganize the graphs and provide *ex vivo* fluorescence images.

Our response: We really appreciate the reviewer for the instructive suggestion. We have performed these experiments and replaced *in vivo* fluorescent images with *ex vivo* fluorescent images, as shown in Figure 3f and 3g, Figure 4c-f and Figure 5a-d of the revised manuscript.

Here is the explanation why the area of fluorescent signal is larger than tumor site. In this work, ICG was used for *in vivo* imaging. Different from other hydrophobic trackers such as DiR, hydrophilic ICG could be well dissolved in aqueous solution and be activated by laser, so the mice presented strong background at initial imaging time points as a result of excitation of ICG in blood circulation. The scale bar was also well organized to present the variation of fluorescent intensity at different time point and differences of distribution among different groups.

3. The RES has two major effects on nanoparticles: blood clearance and tumor accumulation. The authors focused mainly on the tumor accumulation part, while the circulation remains largely uninvestigated. The inclusion of the pharmacokinetic study

would be beneficial for presenting the RES blockade effects.

Our response: We really appreciate the reviewer for the instructive suggestion. We have investigated pharmacokinetic of ICG-loaded 2%NGs without RES-blockade, with 2%-blockade and with 15%-blockade. The data was presented in **Figure S15** and the related text was added to the revised manuscript on Page 15.

In addition to enhanced tumor accumulation, the influence on blood clearance of nanomedicine was another important parameter of RES-blockade. Similarly, 2%-blockade induced negligible improvement in pharmacokinetic of ICG-loaded 2%NGs. In contrast, RES-blockade by 15%NGs significantly prolonged blood half-life of ICG-loaded 2%NGs from 0.699 ± 0.140 h to 2.288 ± 0.358 h, increased AUC from 8.902 ± 2.147 $\text{mg} \cdot \text{L}^{-1} \cdot \text{h}^{-1}$ to 41.493 ± 8.706 $\text{mg} \cdot \text{L}^{-1} \cdot \text{h}^{-1}$, and decreased plasma clearance from 0.648 ± 0.142 $\text{L} \cdot \text{kg}^{-1} \cdot \text{h}^{-1}$ to 0.140 ± 0.032 $\text{L} \cdot \text{kg}^{-1} \cdot \text{h}^{-1}$, as a result of inhibited liver clearance (Fig S15 and Table S4).

4. Figure 1. Captions in the figure are required for a better illustration of what the study is mainly about.

Our response: We thank the reviewer for the advice. We have presented detailed illustration in Figure 1. captions.

Nanogels with distinctive stiffness are injected preferentially and soft nanogels accumulate more in tumor, while stiff nanogels accumulate more in liver and temporarily block RES. Afterwards, therapeutic nanoparticles such as soft DOX-loaded nanogels, commercialized nanomedicine Doxil and Abraxane are injected and antitumor efficacy is obviously promoted after RES-blockade by stiff nanogels.

5. For cell viability tests, incubation with a longer time than 24h is suggested for better showing the safety of the nanogel.

Our response: We really appreciate the reviewer for the suggestion. We have investigated the safety of nanogels for 48 h and 72 h, and the data has been presented in **Figure S3**.

6. For quantitative analysis of cellular uptake, flow cytometry graphs and data should be provided.

Our response: We really appreciate the reviewer for pointing out the problem. We have added flow cytometry graphs and mean fluorescent intensity data to Figure S4, Figure S13, Figure S16, Figure S18 and Figure S21.

7. For the biodistribution study, it is shown that, during 1~4h, the nanogels also distribute significantly in other organs, it would be interesting to investigate where the nanogels accumulate instead of the liver and tumor.

Our response: We really appreciate the reviewer for raising this issue. We have proved soft nanogels accumulated more in tumor and less in liver, while stiff nanogels accumulated more in liver and less in tumor. According to this suggestion, we investigated whether nanogels with different stiffness presented varied accumulation in different organs. In addition of tumor and liver, nanogels also accumulated in lung which contained abundant blood vessels and kidney which was another organ for blood clearance, especially during the first 4 hours after injection (Fig. R1a-R1c). Overall, 15%NGs likely presented a little higher accumulation in lung than 2%NGs. As for kidney, no significant difference could be observed during first 8 hours, except for the 24 h point. Similarly, slightly higher accumulation of 15%NGs could be observed in heart and spleen.

Furthermore, we investigated whether RES-blockade could also affect distribution of ICG-loaded 2%NGs in other major organs. *Ex vivo* images of organs and tumor were captured at 24 h post injection of ICG-loaded 2%NGs with different time interval including 0.5 h, 1.5 h, 3 h, between RES-blockade and drug administration. The results presented similar accumulation of ICG-loaded 2%NGs in tumor and major organs because of weak RES-blockade efficacy of soft 2%NGs (Fig. R2a and R2b). As for RES-blockade by stiff 15%NGs, we could observe enhanced accumulation in tumor and some major organs including lungs and kidneys, because of weakened and delayed clearance by liver (Fig. R2c and R2d).

Figure R1. Stiffness-dependent biodistribution. **a** Biodistribution of ICG-loaded nanogels with different stiffness. **b-e** Semi-quantification of the quantity of ICG-loaded nanogels with different stiffness in organs. Error bars represent standard deviation ($n = 4$ biological independent replicates). Statistical significance was calculated by t -test. P -values: *, $P < 0.05$; **, $P < 0.01$; ***, $P < 0.001$; ****, $P < 0.0001$; ns stands for not significant.

Figure R2. Biodistribution after RES-blockade. **a** Biodistribution of ICG-loaded 2%NGs at 24 h after RES-blockade by 2%NGs with different time interval (0.5 h, 1.5 h, 3h) between RES-blockade and drug administration. **b** Semi-quantification of the quantity of ICG-loaded 2%NGs in tumor and organs. Error bars represent standard deviation ($n = 3$ biological independent replicates). Statistical significance was calculated by one-way ANOVA. P -values: *, $P < 0.05$; **, $P < 0.01$; ***, $P < 0.001$; ****, $P < 0.0001$. ns stands for not significant. **c** Biodistribution of ICG-loaded 2%NGs at 24 h after RES-blockade by 15%NGs with different time interval (0.5 h, 1.5 h, 3h) between RES-blockade and drug administration. **d** Semi-quantification of the quantity of ICG-loaded 2%NGs in tumor and organs. Error bars represent standard deviation ($n = 3$ biological independent replicates). Statistical significance was calculated by one-way ANOVA. P -values: *, $P < 0.05$; **, $P < 0.01$; ***, $P < 0.001$; ****, $P < 0.0001$. ns stands for not significant.

8. In the Introduction section, when mentioning the modification method, the authors need to cite appropriate evidence for the statement: “potential immune suppress induced by CD47-derived peptide will cause unexpected tumor growth or metastasis”.

Our response: We really appreciate the reviewer for raising this issue. Many works have proven the close correlation among CD47, immunosuppression, and tumor growth and metastasis. CD47 is a kind of transmembrane protein expressed on many cancer cells, combining with signal regulatory protein- α expressed on macrophages and inhibiting phagocytosis by macrophages (Clinical Cancer Research 2015, 21: 2325-2337; Cell 2009, 138: 271-285). High expression of CD47 has been proven to correlate positively with tumor invasion and metastasis of non-small cell lung cancer (Scientific Reports 2016, 6: 29719). And CD47 promotes the stemness and malignancy of colorectal cancer, associated with epithelial-mesenchymal transition (Pathobiology 2019, 86: 182-189). In bone metastasis models, less bone tumor burden and tumor-associated bone destruction can be observed in CD47^{-/-} mice, compared to wild-type mice (Cancer Research 2009, 69: 3196-204). CD47 is required for dissemination of non-Hodgkin lymphoma to major organs while CD47 antibodies can inhibit this dissemination progress (Blood 2011, 118: 18). Moreover, blocking CD47 with antibodies can suppress growth of osteosarcoma (Oncotarget 2015, 6: 27). So, we analyzed and concluded that potential immune suppression induced by CD47-derived peptide will cause unexpected tumor growth or metastasis.

Reviewer #2: This manuscript is addressing a strategy to change biodistribution of intravenously injected nanoparticles by inhibiting nanoparticle clearance through macrophage blockade. To inhibit the clearance, saturation of phagocytosis was induced by prompting macrophage endocytosis of stiff nanogels. Using this strategy, authors achieved redistribution of soft nanogels loaded with anticancer drug, which then led to improvement of anticancer effect in mouse model. Applicability of approach for FDA-approved nanoparticles was also demonstrated. While the overall concept of RES blockade is interesting and appealing to researchers working in the nanomedicine space, there are several concerns with this study:

Our response: We thank this reviewer for the positive evaluation and we will try our best to address the concerns.

Major comments:

1. Drug loading can change colloidal stability of nanoparticles, as well as other their properties. Authors investigated this issue only for 2% NGs. The same studies must be performed for 15% NGs, which is widely used in the manuscript.

Our response: We really appreciate the reviewer for raising this issue. We have characterized and compared parameters of 15%NGs and DOX@15%NGs, and no obvious variation could be observed after loading DOX. Relevant data was presented in **Figure S2**.

2. Treatments with nanogels lead to a drop of mice weight by several grams, with no returning for normal values. It indicates toxicity, which should be explained and described. Nevertheless, authors mentioned that only free dox have cardiotoxicity, refereeing on Fig 3L, which is not obvious from this graph. I should add that the same weight loss can be observed in Figure 6E.

Our response: We really appreciate the reviewer for pointing out the problem. We have corrected our description and discussed the possible reason for loss of body weight in revised manuscript on Page 14.

The body weight showed temporary decrease in G3 and G4, which might be caused by higher accumulation in major organs, and recovered over time after drug administration

(Fig. 3k). Meanwhile, H&E staining exhibited no obvious toxicity in major organs after treatment (Fig. S10).

And on Page 24.

Slight loss of body weight could be observed during treatment which might be caused by higher accumulation of DOX-loaded nanogels in major organs, however, the body weight recovered to normal level after treatment and negligible toxicity toward major organs could be observed in H&E staining at the end of experiment (Fig. 6e and S25).

And on Page 27.

Slight decrease of body weight in G4 might result from delayed clearance of Doxil after RES-blockade by 15%NGs (Fig 7e). No obvious toxicity toward major organs was detected in H&E staining (Fig. S29).

3. In blood biochemical parameters, several indicators significantly decreased (see, for example, WBC: Figures S19, S23). Even if they are in normal range, these accidents should be mentioned in the text and discussed, as possible signs of toxicity.

Our response: We really appreciate the reviewer for pointing out the problem. We have improved our statement and provided explanation for the variation of some indicators. As for WBC, we consider the decrease was not caused by treatment, instead it was correlated with tumor growth and malignancy. It has been reported that inflammation occurred during tumor development and the level of WBC also increased consistently (Nature 2008, 454: 436-444; British Journal of Cancer 2007, 97: 1266-1270; Ling Cancer 2014, 85: 457-464).

Here is the added discussion on Page 14.

In blood biochemical analysis and blood routine examine, the variation of white blood cells (WBC) could be observed and the number of WBC were always positively correlated with tumor volume, which could be a result of tumor malignancy and inflammation, while all the physiological indicators were still in the normal range (Fig. S11).

And on Page 24.

In blood biochemical analysis and blood routine examine, in addition of the varied WBC level, the ALT level obviously increased in G7 might result from burden on liver biodegrading and clearing 15%NGs. Meanwhile, all the physiological indicators were still

in the normal range (Fig. S26).

And on Page 27.

Relatively higher ALT level was examined in G4 and G7, which were blocked by 15%NGs, similar to Figure S26a, in blood biochemical analysis and blood routine examine (Fig. S30). However, all the indicators were in normal range and exhibited that commercialized nanomedicine combined with 15%-blockade caused negligible toxicity and side effects.

4. Time interval for RES blockade *in vivo* was determined from the *in vitro* experiments. Since nanoparticle behavior in mice bloodstream is more complex than interaction of nanoparticles with macrophages *in vitro*, this study raises serious concerns: What is the blood circulation time of 2% and 15% NGs *in vivo*? Why was the cellular uptake determined only for 15% NGs? Can the time interval be different for nanogels with different stiffness? Authors should rationalize their injection regime by experiments on mice.

Our response: We really appreciate the reviewer for your instructive comments.

First, we investigated pharmacokinetic of nanogels with different stiffness. The result showed that both soft 2%NGs and stiff 15% nanogels presented relatively short blood half-life (Figure R3). After calculating, we found that both nanogels showed similar blood half-life, while $AUC_{(0-t)}$ of 15%NGs was over 1.86-fold of 2%NGs (Table R1). However, we could observe that 2%NGs presented quite lower percentage of injected dose ($14.1 \pm 3.3\%$ I.D.) compared to 15%NGs ($41.9 \pm 2.0\%$ I.D.) at 5 min post injection (Figure R3). In combination with analysis of *ex vivo* and *in vivo* imaging (Figure 3f, 3g and S5), which presented decrease of fluorescent intensity over time instead of continuous increase, we concluded that rapid biodistribution occurred on nanogels after injection, and biodistribution of soft 2%NGs was even faster.

Figure R3. Pharmacokinetic of nanogels with different stiffness. Error bars represent standard deviation ($n = 3$ biological independent replicates).

Table R1. Pharmacokinetic parameters of nanogels with different stiffness.

	$t_{1/2}$ (h)	AUC ($\text{mg}\cdot\text{L}^{-1}\cdot\text{h}^{-1}$)	Vd ($\text{L}\cdot\text{kg}^{-1}$)	CL ($\text{L}\cdot\text{kg}^{-1}\cdot\text{h}^{-1}$)
2%NGs	0.622 ± 0.213	7.322 ± 3.418	0.705 ± 0.099	0.898 ± 0.489
15%NGs	0.355 ± 0.079	13.610 ± 2.855	0.197 ± 0.006	0.396 ± 0.073

Second, the reason why the cellular uptake determined only for 15% NGs was that 15%NGs accumulated much more in liver than 2%NGs and we speculated 15%NGs was more appropriate for RES-blockade. The following experiment confirmed our speculation that 15%NGs presented excellent RES-blockade efficacy while quite weak RES-blockade effect could be observed for 2%NGs (Fig. 4). So, the time-dependent cellular uptake of 2%NGs was not further taken into consideration.

Of course, only cell experiments could not completely verify that 1.5 h was the optimal time interval for soft 2%NGs and stiff 15%NGs used for RES-blockade. In vivo imaging of ICG-loaded 2%NGs after 2%-blockade or 15%-blockade with different time interval including 0.5 h, 1.5 h, 3 h was investigated. And the results showed that all the 2%-blockade groups didn't promote tumor accumulation or inhibit tumor accumulation, while all the 15%-blockade groups increased accumulation of ICG-loaded 2%NGs in tumor and decreased accumulation in liver (Figure R4a and R4d). However, time interval between RES-blockade

and drug administration didn't present obvious influence on RES-blockade efficiency, both in 2%-blockade groups and 15%-blockade groups (Figure R4b and R4c, R4e and R4f).

Figure R4. Pharmacokinetic of nanogels with different stiffness. **a** Tumor accumulation and liver accumulation of ICG-loaded 2%NGs after RES-blockade by 2%NGs with different time interval. Semi-quantification of the quantity of ICG-loaded 2%NGs in **b** tumor and **c** liver. Error bars represent standard deviation ($n = 4$ biological independent replicates). Statistical significance was calculated by one-way ANOVA. P -values: *, $P < 0.05$; **, $P < 0.01$; ***, $P < 0.001$; ****, $P < 0.0001$. ns stands for not significant. **d** Tumor accumulation and liver accumulation of ICG-loaded 2%NGs after RES-blockade

by 15%NGs with different time interval. Semi-quantification of the quantity of ICG-loaded 2%NGs in **e** tumor and **f** liver. Error bars represent standard deviation ($n = 4$ biological independent replicates). Statistical significance was calculated by one-way ANOVA. P -values: *, $P < 0.05$; **, $P < 0.01$; ***, $P < 0.001$; ****, $P < 0.0001$. ns stands for not significant.

5. Analysis of variance (ANOVA) is preferable for statistical analyses between three or more groups to avoid type 1 error.

Our response: We really appreciate the reviewer for this valuable suggestion. One-way ANOVA has been applied to appropriate data and illustrated in relative figure captions.

6. Authors claimed suppression of clathrin-mediated endocytosis as a major mechanism of blockade. Nevertheless, this mechanism was defined from *in vitro* studies and can be non-sole *in vivo*. Authors should increase discussion about other possible mechanism which can cause observed blockade.

Our response: We really appreciate the reviewer for the advice. We investigated the correlation between clathrin-mediated endocytosis and RES-blockade because we found that nanogels blocked macrophages was not simply by saturation as a conclusion of stiff 15%NGs blocking RES with lower dosage than soft 2%NGs. So, we speculated that the internalized function of macrophages was inhibited and found that clathrin-mediated endocytosis was suppressed by stiff 15%NGs. And this was the *in vitro* mechanism of RES-blockade at cellular level (Fig. 5f, 5g and S22). On the other hand, another *in vivo* mechanism was that stiff 15%NGs was harder to pass through liver sinusoidal fenestration relying on lower deformability and possessed of more chance to be internalized by macrophages than soft 2%NGs. In addition, dense crosslinking structure made 15%NGs hard to be degraded and prolonged retention time of 15%NGs in liver (Fig. 5h and 5i). Overall, inhibition of clathrin-mediated endocytosis of macrophages, as well as higher accumulation and prolonged retention in liver should be the mechanism of RES-blockade.

Minor comments:

1. Line 45-46: Add than in average, 0.7% was delivered.

Our response: We really appreciate the reviewer for pointing out the problem. The statement has been corrected to be more precise on Page 3.

Although pharmacokinetics, antitumor efficacy and safety of nanomedicines have achieved great benefits, only 0.7% of nanoparticles in average can be delivered to solid tumors.

2. Lines 49-50: Authors suppose that RES = MPS, which is not correct since these systems include different types of cells. They should clarify which system they blockade – if it only macrophages, it rather MPS blockade then RES blockade.

Our response: We really appreciate the reviewer for pointing out the problem. It was not a rigorous statement that RES was also referred to as MPS, and we have adjusted our introduction on Page 3.

Surface of nanoparticles is rapidly covered with complex serum proteins after *i.v.* injection, resulting in recognition and quick clearance from blood circulation by macrophages in reticuloendothelial system (RES).

According to our *in vivo* imaging data (Fig. S14 and S17), liver presented a quite strong fluorescent intensity after injection, which meant large amount of nanomedicine accumulated in liver. And macrophages in liver played a significant role in clearance of nanomedicine from blood circulation. Liver and macrophage are important components of reticuloendothelial system, and we have tried to explain the mechanism of blockade from two aspects including inhibition of macrophage internalization as well as high accumulation and prolonged retention of 15%NGs in liver. So, we think it is logical to intitule as RES-blockade strategy. Without doubt, liver is a complicated organ composed of different cells including endothelial cell, stellate cell and hepatocyte (Journal of controlled release 2016, 240: 332-348). It needs further investigation to understand whether nanogels with different stiffness also affect other cells.

3. Introduction: a lot of recent achievements in the macrophage blockade field were ignored, like nanoprimers technology (Saunders et al, Nano letters, 2020); cytoblockade with RBCs (Nikitin et al, Nat. Biomed. Eng., 2020) or exosomes blockade (Belhadj, J. extracellular vesicles, 2020).

Our response: We really appreciate the reviewer for the careful review. We have added related description and introduction in Page 3 and 4.

MPS-erythroblockade, which is utilizing a low dose of allogeneic anti-erythrocyte antibodies and forcing MPS to clear erythrocyte, has been proven to increase the blood circulation half-life of nanoparticle formulations. Pre-injecting cationized mannan-modified extracellular vesicles and then injecting drug-loaded nanocarriers fused with CD47-enriched exosomes, lead to prolonged circulation time and increased tumor accumulation.

4. Lines 253-255: specify is it mean tumor volumes? Add SD

Our response: We really appreciate the reviewer for pointing out the problem. The statement has been revised to be more precise. And all SD has been added after the mean value.

Better antitumor efficacy was obtained by DOX@10%NGs with tumor volume of around $394.1 \pm 66.7 \text{ mm}^3$ and the best was by DOX@2%NGs with around $297.1 \pm 78.3 \text{ mm}^3$ relying on better permeability ascribed to excellent deformability of soft nanogels (Fig. 3j).

5. Fig 4a, c, d and other bioimaging – add quantitative scale diapason

Our response: We really appreciate the reviewer for pointing out the problem. All quantitative scale diapason of bioimaging images has been added.

6. Fig S13 – correct typo “particals”

Our response: We really appreciate the reviewer for pointing out the mistake. Similar mistakes have been corrected in revised manuscript.

Point-by-point response to the reviewers' comments

We really appreciate all reviewers for their rigorous and meticulous comments. Below, the comments are shown in blue, and our responses are in standard typeface. Sections highlighted in yellow represent wording that has been altered/inserted into the revised manuscript. The corresponding sections in the revised manuscript itself are also highlighted in yellow.

Reviewer #1: The manuscript is thoroughly revised by including additional justifications and supplementary data. There remains, however, some questions that needs to be addressed prior to publication. With appropriate revisions in the manuscripts on these aspects, the manuscript may be ready for publication.

Our response: We thank this reviewer for the careful review. We will try our best to revise the manuscript.

1. Shrinkage of nanogels during the TEM measurements can't allow to compare their sizes, so the phrase in lines 174-176 should be corrected. It can be reasonable to leave only AFM data in the text, or comment shrinkage in the result description.

Our response: We really appreciate the reviewer for pointing out the problem. We have deleted the description "close diameters" in the manuscript, which demonstrated similar meaning to "monodispersed", to avoid ambiguity of expression. Besides, we added explanation of difference in diameter of different samples in TEM images on Page 8.

Slight difference in diameter among TEM images resulted from shrinkage of nanogel network structure after drying during sample preparation of TEM samples.

2. Figure R1-R2 are not well correlate with the data in the manuscript.
 - 2a. First, in Figure R1 2% NGs showed decrease of the liver uptake, but there was no any increase of accumulation in other organs. Did you analyze accumulation in the tumor in this experiment?
 - 2b. Second, in Figure R2 RES blockade with 2% NGs show not only decrease uptake of ICG-labeled nanoparticles in tumor, but also in liver. It contradicts with Figure 4 of

the main manuscript. In addition, total radiant efficiency of 2% NGs is much lower than for 15% NHs.

2c. Data of biodistribution should be presented in manuscript, at least in Supplementary Materials.

Our response: We really appreciate the reviewer for the careful review.

2a. Tumor accumulation of ICG-loaded nanogels with different stiffness was presented in Figure 3f and 3g. It could be observed that 2%NGs exhibited increased accumulation in tumor and less uptake by liver, compared to 15%NGs.

2b. There are two main reasons for the appearance of lower accumulation of ICG-loaded 2%NGs both in tumor and liver after RES-blockade by 2%NGs (Fig. R2a and R2b), compared to RES-blockade by 15%NGs (Fig. R2c and R2d).

First, the two experiments, RES-blockade by 2%NGs (Fig. R2a and R2b) and RES-blockade by 15%NGs (Fig. R2c and R2d), were proceeded in different batches, including different batches of tumor models and varied physiological status of mice which might affect elimination of nanogels.

Second, the tumor and organs were harvested at 24 h after drug administration of ICG-loaded 2%NGs. Most nanogels have been eliminated from liver and the fluorescent intensity in liver was only 20% of 1 h.

So, the difference was generated between the two experiments. It was not precise to compare the fluorescent intensity between the two experiments (Fig. R2a, R2b and Fig. R2c, R2d), instead, comparison of experimental groups with control groups of each experiment was more significative. The observation of lower total fluorescent intensity in experiment after RES-blockade by 2%NGs (Fig. R2a and R2b) also resulted from varied physiological status and elimination in different batches of tumor models.

2c. Fluorescent images of Figure R1 (Fig. R1a) and Figure R4 were added to supplementary materials (Fig. S5 and Fig. S15) and relevant description was added to manuscript on Page 13.

The result demonstrated that 2%NGs could achieve higher accumulation at tumor site than 15%NGs (Fig. 3f and 3g) and accumulation of nanogels in major organs was presented in Fig. S5.

And on Page 15.

However, *in vivo* imaging of ICG-loaded 2%NGs with different time interval from 0.5 h to 3 h presented no obvious differences, both in 2%-blockade and 15%-blockade groups (Fig. S15). In consideration of saturating macrophages and no more increase of tumor accumulation with prolonged time interval, 1.5 h was set as the time interval between RES-blockade and subsequent drug administration, in consistence with previous work

Quantification data of liver in Figure R1 has been presented in Figure 5i, so this part of data was excluded (Fig. R1b-e). To avoid misleading readers, we considered it was not appropriate to present Figure R2 in the manuscript. In addition, influence from time interval between RES-blockade and drug administration has been excluded by Figure R4, so Figure R2 was not presented in the revised manuscript.

3. Figure S15 present several circulation kinetics with elimination until equal 4% baseline. Is it NGs or released ICG circulate in the blood? How did you measure “0 concentration” level? Baseline level can greatly influence AUC values.

Our response: We really appreciate the reviewer for raising this issue.

The remained around 2% of injected dosage contained both ICG-loaded 2%NGs and little released ICG because of quick elimination of small molecules by kidney.

The “0 concentration” or the background of samples was obtained by detecting the fluorescent intensity of the mixture of 10 μ L of plasma without drug administration and 50 μ L of DMSO, consistent to the preparation of experimental samples.

The reason why fluorescent signal could still be detected at 12 h should be the remained nanogels in blood circulation. Further decrease of fluorescent intensity might be observed after prolonged detection.

The AUC values could be affected by the baseline, while in this work we focused on the point how the RES-blockade by nanogels with different stiffness affected pharmacokinetics, and the variation of pharmacokinetics or conclusion were not affected by the baseline.

4. Figure S15 caption: Correct “Pharmacokinetics”

Our response: We really appreciate the reviewer for pointing out the mistake. We have

corrected the mistakes in the manuscript and supplementary materials.

5. Please, add measurement units in “radiant efficiency” scales

Our response: We really appreciate the reviewer for pointing out the problem. We have added measurement unit of fluorescent intensity in the figures of manuscript and supplementary materials.

Reviewer #2: Authors significantly improved the paper and I satisfied with their answers. Only several minor corrections should be made. After the revision paper can be published even without reevaluation of the manuscript.

Our response: We really appreciate this reviewer for the instructive suggestion. We will try our best to further improve the manuscript.

1. Line 55: “it causes another trouble that is restrained binding and limited internalization by tumor cells” – please cite appropriate reference.

Our response: We really appreciate the reviewer for pointing out the problem. Relevant references have been cited.

2. Describe terms after first mention: MPS, DOX, P(NIPMAM-ss-MAA), GSH

Our response: We really appreciate the reviewer for figuring out the issue. Describe terms of these and other abbreviations have been added when they were first mentioned.

3. Line 279: “However, short wavelength of Rhodamine” – specify. Is it absorption or fluorescence wavelength?

Our response: We really appreciate the reviewer for the careful review. Fluorescent excitation and emission wavelength of Rhodamine B have been illustrated on Page 15.

However, short fluorescent wavelength ($\lambda_{\text{ex}} = 568 \text{ nm}$, $\lambda_{\text{em}} = 583 \text{ nm}$) of Rhodamine led to dissatisfactory *in vivo* imaging effect, the mice were sacrificed and tumors were harvested for *ex vivo* imaging 4 h post-injection (Fig. S7a).

4. All the measurements report exact values with many decimal places more than what is seems realistic. To simplify data comparison and paper reading some numbers can be rounded. For example, if the tumor volume is 492.5 ± 109.5 , SD is too high for such accuracy, it can be rounded to 493 ± 110 .

Our response: We really appreciate the reviewer for the instructive suggestion. Some data which was too accurate has been rounded to make results more definite and logical.

5. Line 364 – typo “EX vivo”

Our response: We really appreciate the reviewer for pointing out the mistake. The mistake has been corrected in figure caption of Figure 4.

6. Line 499 – “In contrast, 15%NGs was stiff enough to withstand biodegradation and elimination by liver and therefore sustained for a longer time, leading to long-lasting RES-blockade.” -I am not sure the nanoparticles, which already was transported to the lysosomes for the biodegradation still can blockade macrophages and cause “long lasting effects”. Stiff nanogels induce more efficient RES blockade, but there is no data about effect duration. I suppose this one and similar phrases should be corrected.

Our response: We really appreciate the reviewer for raising this issue. In this work, we concluded that stiff nanogels achieved RES-blockade effect by inhibiting internalization of macrophages. We considered 15%NGs was harder to biodegrade and eliminate, and 15%NGs presented longer retention in liver (Fig. 5h and 5i), so they could affect macrophages for a longer time. Although around half of 15%NGs were eliminated from liver at 3 h after injection (Fig. 5h and 5i), 15%NGs could still achieve similar RES-blockade efficiency to time interval of 0.5 h or 1.5 h between RES-blockade and drug administration (Fig. S15), which also meant function of macrophages was inhibited. So, we analyzed that prolonged retention of 15%NGs in liver could continuously inhibit internalization of macrophages and achieve sustained RES-blockade. We thought it was reasonable. Certainly, this result is only a possible reason for 15%NGs achieving better RES-blockade than 2%NGs, and these data is not sufficient to give a clarified or complete illustration of the relationship between liver retention and RES-blockade. More detailed experiments

should be performed in the following study.

7. There is no need to repeat long phrases about statistical methods in each Figure letter. Maybe It will be better to include it in the end of the caption? For example: "In b,c,f,g P-values: *, $P < 0.05$; **, $P < 0.01$; ***, $P < 0.001$; ****, $P < 0.0001$. ns stands for not significant."

Our response: We really appreciate the reviewer for the instructive advice. We have simplified the figure captions of all Figures.

Reviewers' Comments:

Reviewer #2:

Remarks to the Author:

I am satisfied with the author's responses to the questions.
The manuscript can be published.

Reviewers' Comments:

Reviewer #2:

Remarks to the Author:

I am satisfied with the author's responses to the questions.
The manuscript can be published.

Point-by-point response to the reviewers' comments

We really appreciate all reviewers for their rigorous and meticulous comments. Below, the comments are shown in blue, and our responses are in standard typeface.

Reviewer #2: I am satisfied with the author's responses to the questions.

The manuscript can be published.

Our response: We really appreciate this reviewer for the positive evaluation.